# Dynamics of the compartmentalized *Streptomyces* chromosome during metabolic differentiation

Virginia S. Lioy [1✉], Jean-Noël Lorenzi [1], Soumaya Najah [1], Thibault Poinsignon [1], Hervé Leh[1], Corinne Saulnier[1], Bertrand Aigle [2], Sylvie Lautru[1], Annabelle Thibessard [2], Olivier Lespinet[1], Pierre Leblond[2], Yan Jaszczyszyn[1], Kevin Gorrichon[1], Nelle Varoquaux[3], Ivan Junier[3], Frédéric Boccard[1], Jean-Luc Pernodet [1] & Stéphanie Bury-Moné [1✉]

Bacteria of the genus *Streptomyces* are prolific producers of specialized metabolites, including antibiotics. The linear chromosome includes a central region harboring core genes, as well as extremities enriched in specialized metabolite biosynthetic gene clusters. Here, we show that chromosome structure in *Streptomyces ambofaciens* correlates with genetic compartmentalization during exponential phase. Conserved, large and highly transcribed genes form boundaries that segment the central part of the chromosome into domains, whereas the terminal ends tend to be transcriptionally quiescent compartments with different structural features. The onset of metabolic differentiation is accompanied by a rearrangement of chromosome architecture, from a rather 'open' to a 'closed' conformation, in which highly expressed specialized metabolite biosynthetic genes form new boundaries. Thus, our results indicate that the linear chromosome of *S. ambofaciens* is partitioned into structurally distinct entities, suggesting a link between chromosome folding, gene expression and genome evolution.

[1] Université Paris-Saclay, CEA, CNRS, Institute for Integrative Biology of the Cell (I2BC), Gif-sur-Yvette, France. [2] Université de Lorraine, INRAE, DynAMic, Nancy, France. [3] Université Grenoble Alpes, CNRS, Grenoble INP, TIMC-IMAG, Grenoble, France. ✉email: virginia.lioy@i2bc.paris-saclay.fr; stephanie.bury-mone@i2bc.paris-saclay.fr

Bacteria of the genus *Streptomyces* are amongst the most prolific producers of specialized metabolites with applications in medicine, agriculture, and the food industry[1]. The biosynthesis of these specialized metabolites (e.g. antibiotics and pigments) generally occurs after 'metabolic differentiation', a physiological transition from primary to specialized metabolism[2,3]. This transition coincides with morphological differentiation[4] (i.e. formation of the secondary multinucleated mycelium, sporulation) that occurs late in the growth phase. These metabolic and morphological changes are controlled by highly interconnected regulatory mechanisms that have been studied mostly at the transcriptional, translational, and post-translational levels[4–7]. However, a clear link between global chromosome organization and metabolic differentiation has yet to be established.

*Streptomyces* possess an unusual linear chromosome. Moreover, its extremities, or 'terminal arms', contain terminal inverted repeats (TIRs) capped by telomere-like sequences. In addition, the chromosome is large (6–15 Mb) with an extreme GC content (circa 72%). Finally, the *Streptomyces* chromosome presents remarkable genetic compartmentalization, with a distinguishable central region and the terminal arms. The central region primarily contains core genes, common to all *Streptomyces* species, these being often essential[8–13]. The terminal arms are mainly composed of conditionally adaptive genes, in particular, enriched in specialized metabolite biosynthetic gene clusters (SMBGCs)[14]. In addition, they are prone to large DNA rearrangements and frequent recombination[12,15–19]. Importantly, many SMGBCs appear silent or poorly expressed under laboratory conditions, giving rise to the concept of 'cryptic' SMBGCs. The basis of the regulation of these clusters in relation to the overall genome dynamics remains to be explored.

The development of chromosome conformation capture (3C) methods coupled to deep sequencing provided novel concepts in bacterial chromosome biology[20,21]. Notably, pioneer studies revealed that bacterial chromosomes present a high degree of 3D-organization in macrodomains and/or 'chromosome interacting domains' (CIDs), mediated by multiple structural factors, including transcription and replication[22–29]. In this work, we explore the dynamics of *Streptomyces ambofaciens* ATCC 23877 linear chromosome during metabolic differentiation. This strain, well-studied for its genome organization and plasticity[12,16,17,30], is industrially exploited for the production of spiramycin. By combining multi-omics approaches, we show that the dynamics of gene expression correlate with the folding of the linear chromosome of *S. ambofaciens* ATCC 23877 into transcriptionally active and silent compartments. Moreover, metabolic differentiation is accompanied by a huge rearrangement of chromosome architecture from a rather 'open' to 'closed' conformation, in which SMBGCs form new boundaries. Altogether, our results highlight a link between chromosome folding, gene expression and genome evolution.

## Results

### Genetic compartmentalization of the *S. ambofaciens* genome.

The genetic compartmentalization of the *S. ambofaciens* ATCC 23877 genome was previously reported[12]. However, here we took advantage of the numerous available *Streptomyces* sequences to update the cartography of the genome. We also used gene persistence[31] as an indicator to describe the organization of the *Streptomyces* genome. Gene persistence reflects the tendency for genes to be conserved in a large number of genomes. As shown in other bacteria, gene persistence is associated with gene essentiality and similar expression levels[31–33]. Here, we calculated a gene persistence index by determining the frequency of a given gene in a panel of 125 complete *Streptomyces* genomes (Supplementary Table 1)[30]. The highest level of persistence associated with the best reciprocal matches between genes of all these genomes defines the core-genome[30]. In addition, we mapped the genes encoding the 'actinobacterial signature' previously identified[34] as those coding sequences (CDSs) that are nearly universal among actinobacteria. In contrast, we searched the *S. ambofaciens* ATCC 23877 genome for variable regions, potentially acquired by horizontal gene transfer. For this purpose, we identified unique genes as well as genomic islands (GIs, Supplementary Data 1, Supplementary Fig. 1). These are defined as DNA sequences that are inserted in regions of synteny (i.e. same orthologous gene order in closely related genomes, see the "Methods" section). Thus genes from the core, the actinobacterial signature, and/or presenting a high level of persistence are enriched in the central region, whereas GIs and unique CDSs are enriched in chromosome extremities. The results clearly highlight the compartmentalization of the *S. ambofaciens* genome (Fig. 1, Supplementary Table 2). Synteny gradually disappears in terminal arms (Fig. 1), as previously reported[12]. This makes it difficult to have an operational delineation of the limits of these arms. However, since the first and last ribosomal operons approximately mark the limits of a central region beyond which the synteny level falls (Fig. 1), these operons were used as limits for the definition of the left and right extremities in this study.

The functional annotation also highlighted a bias of gene distribution along the chromosome (Fig. 1, Supplementary Table 2). As a proxy for major metabolic processes, we used the functional RNA genes (encoding rRNA, tRNA, tmRNA, RNAse-P RNA, SRP-RNA) and genes encoding functions related to translation and/or RNA stability. We also mapped the genes encoding nucleoid-associated proteins (NAPs) and chromatin structural factors, further referred to as 'NAPSFs', that play a central role in the dynamic organization of the bacterial chromosome and are enriched in the central region (Fig. 1, Supplementary Table 2). Finally, we used the antiSMASH secondary metabolite genome mining pipeline (v5.1.0)[35] to identify putative SMBGCs (Supplementary Data 2). These genes are preferentially located in the terminal arms and almost half of them are present within GIs (Fig. 1, Supplementary Table 2). Accordingly, GIs are 3.5-fold enriched in SMBGC genes ($p$ value < $2.2 \times 10^{-6}$, two-sided Fisher's exact test for count data). These results illustrate that the high variability of *Streptomyces* extremities relates in particular to functions involved in metabolic differentiation.

Together, these analyses confirm the strong genetic compartmentalization of the *S. ambofaciens* chromosome. In this context, we then explored the extent to which this genetic organization correlates with gene expression and chromosome architecture.

### Transcription is compartmentalized in *S. ambofaciens*.

To focus specifically on transcriptome dynamics during metabolic differentiation, while limiting cellular physiological heterogeneity in the colony, we took advantage of the fact that *S. ambofaciens*, like many *Streptomyces*[2], does not sporulate in a liquid medium. We chose MP5 and YEME10 liquid media (Table 1) in which *S. ambofaciens* grows in a rather dispersed manner that makes the cells also more accessible to further 3C-treatments. Moreover, for *S. ambofaciens*, the MP5 medium was previously reported to be suitable for the production of the antibiotics spiramycin[36] and congocidine[37] while there is only limited antibiotic production in the YEME10 medium (Supplementary Fig. 2). The results showed that the transcriptomes were rather similar in exponential phase in both media, while commitment to specialized metabolism (C4, C5, and C7 conditions) was accompanied by major

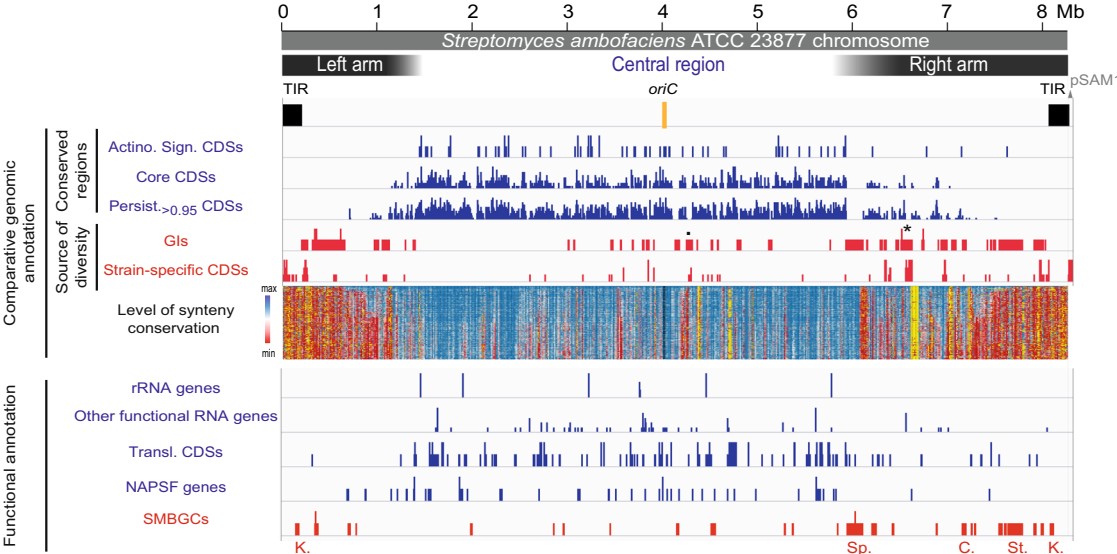

**Fig. 1 The genetic compartmentalization of *S. ambofaciens* linear chromosome.** The terminal inverted repeats (TIRs, 200 kb) and the origin of replication (*oriC*) are indicated. The distribution of gene presenting features of interest are represented in blue or red when enriched in the central region or within the extremities (defined by the first and last rDNA operons), respectively (see Supplementary Table 2 for statistical analysis of the data). The dot and asterisk indicate the position of pSAM2 and a complete prophage, respectively. The position of the SMBGCs encoding the biosynthesis of all known antibacterial compounds of *S. ambofaciens* ATCC 23877 is also indicated. The level of synteny along the chromosome of *S. ambofaciens* ATCC 23877 is represented as a heat map of gene order conservation (GOC) scores using a sliding window (8 CDSs with 1 CDS step). Each line corresponds to the GOC profile of the reference against another species. The 124 species are organized from the phylogenetically closest to the furthest compared to *S. ambofaciens*. The vertical black dotted line represents the location of the *dnaA* gene delineating the two replicators. "NA" (in yellow) indicates the absence of orthologs in the corresponding window. *S. ambofaciens* ATCC 23877 also harbors a circular pSAM1 plasmid (≈89 kb). *Abbreviations*: 'Actino. Sign. CDSs' (coding sequences of the actinobacterial signature); BGC (biosynthetic gene cluster); 'C.' (congocidine BGC); 'Chr.' (whole chromosome); GIs (genes belonging to genomic islands); 'K.' (kinamycin BGC); 'Persist.$_{>0.95}$ CDSs' (coding sequences of *S. ambofaciens* ATCC 23877 presenting a gene persistence superior to 95% in 124 other *Streptomyces* genomes); 'Transl. CDSs' (genes encoding functions involved in translation process and/or RNA stability); NAPSFs (nucleoid-associated proteins and structural factors); SMBGCs (specialized metabolite BGCs); 'Sp.' (spiramycin BGC); 'St.' (stambomycin BGC); 'rRNA' (ribosomal RNA).

**Table 1 Growth conditions used to perform -omics analyses.**

| Conditions | Media | Brief description | Time (h) | Antibacterial activity[a] |
|---|---|---|---|---|
| C1 | MP5 | Liquid medium based on yeast extract containing glycerol (3.6%), NaNO$_3$ (1 g/l), MOPS-optimized for congocidine and spiramycin production | 24 | − |
| C2 | | | 30 | − |
| C3 | | | 36 | − |
| C4 | | | 48 | ++ |
| C5 | | | 72 | +++ |
| C6 | YEME10 | Liquid medium based on yeast extract containing saccharose (10.3%), glucose (1%), and malt extract | 24 | − |
| C7 | | | 48 | + |

[a]Size of the inhibition halo: "−" (no detectable inhibition), "+" (<20 mm), "++" (]20; 30] mm), "+++" (>30 mm) (see Supplementary Fig. 2).

transcriptional changes that form a distinct cluster (Fig. 2a, b, Supplementary Fig. 3). Notably, at the earliest time points in the growth period in both media, the terminal regions were rather poorly expressed, whereas transcription gradually increased toward terminal ends over the growth period (Fig. 2b), as previously reported in *S. coelicolor*[10,38]. We observed that the *S. ambofaciens* genome exhibits a transcriptional landscape with ~90% of the genes significantly expressed in at least one condition (i.e. categorized as 'CAT_1' or higher, in at least one condition—Supplementary Fig. 4a, b, Supplementary Data 3). Moreover, this transcriptome is rather dynamic with more than 93% of the genes differentially expressed in at least one condition (i.e. adjusted *p* value < 0.05 in the DESeq2 differential analysis using the C1 condition as reference, Supplementary Data 3).

As shown by scanning the genome with respect to gene expression levels, we found a marked bias in the localization of poorly versus highly expressed genes along the genome as well as a positive correlation between gene persistence and expression (Fig. 2c, Supplementary Fig. 4c, d). The 332 chromosomal genes expressed at a very high level in all conditions are mainly enriched in the central region of the genome (Supplementary Table 2, Fig. 3d), whereas most (>60%) of the SMBGCs, GIs, and mobile genetic elements (pSAM1, pSAM2, and a prophage) are silent or poorly expressed in at least one condition. Up to 19% and 23% of SMBGC and GI genes, respectively, were found in this category in all tested conditions (Supplementary Fig. 4b). However, compared to other poorly conserved genes, SMBGCs have a higher proportion of highly induced genes, from 'CAT_0'

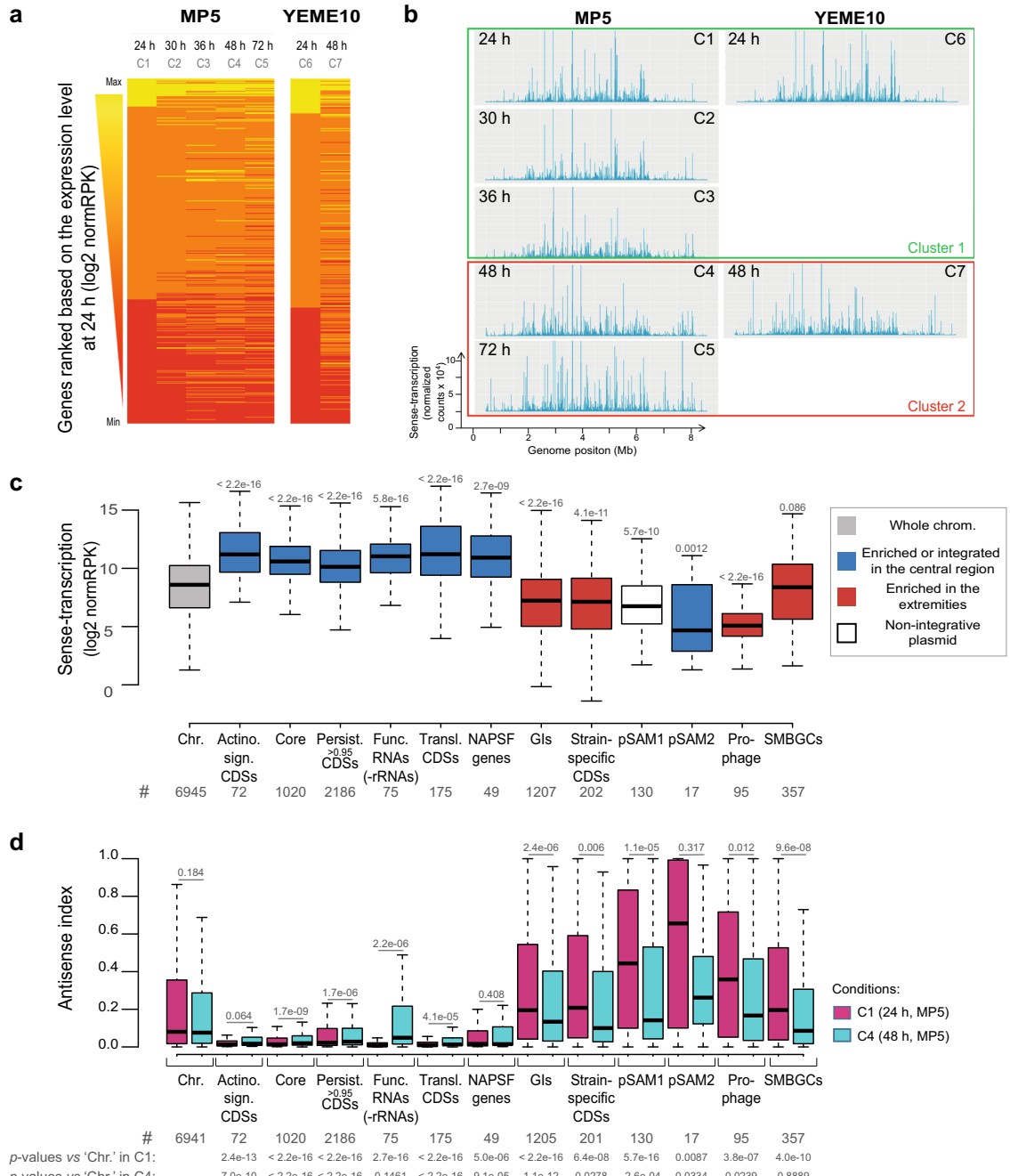

to the highest ('CAT_3' or 'CAT_4') categories (Supplementary Fig. 4c).

Interestingly, as previously described in *S. coelicolor*[38], we observed significant antisense-transcription throughout the genome. Considering the absolute number of transcripts, this antisense-transcription was positively correlated to the level of sense-transcription (Supplementary Fig. 4e, f). Highly transcribed genes promote an environment enriched in transcriptional machinery, which may generate a transcriptional 'leakage' in the antisense orientation. To investigate the relative importance of this antisense-transcription, we defined the 'antisense index' as the level of antisense-transcription over the total (sense plus antisense) transcription. This index is low for genes expressed at a very high level, suggesting that they have developed regulatory mechanisms that increase the probability of localizing correctly the RNA polymerase at the promoter in the sense orientation.

Interestingly, in the exponential phase, the antisense index was particularly high for GIs, mobile genetic elements, and SMBGCs and very low in conserved regions enriched in the central region (except for pSAM2) (Fig. 2d). During metabolic differentiation, the antisense indices of GIs, pSAM1, prophage, and especially SMBGCs tended to decrease (Fig. 2d). This observation suggests that antisense-transcription could be a consequence of or directly involved in the regulation of gene expression concerning metabolic differentiation. In this context, the SMBGC switch-on may rely on mechanisms that limit antisense-transcription by increasing the probability of localizing correctly the transcription machinery at promoters in the sense orientation[39].

Together, these results indicate that the genetic compartmentalization of the *Streptomyces* genome clearly correlates with compartmentalization of transcription, both sense, and antisense.

**Fig. 2 Transcriptome dynamics in *S. ambofaciens* genome. a** Transcriptome changes overgrowth. Heatmaps of the DESeq2 normalized number of reads per kb (normRPK) per gene, ranked by their expression level at 24 h. Each line represents a gene. **b** Transcription along the chromosome overgrowth. The two clusters to which the different conditions belong were obtained by hierarchical classification (Supplementary Fig. 3a). Whereas cluster 1 corresponds to exponential phases (no antibiotic production), cluster 2 gathers conditions associated with the activation of SMBGC encoding at least two known antibacterial activities (Supplementary Fig. 3c). 'C1'–'C7' refers to the name of the studied conditions (see Table 1, for details). **c** Boxplot presenting the mean level of gene expression depending on features of interest. The distribution of the normalized number of reads per kb (normRPK) as the mean of all studied conditions is presented depending on the genome features of interest (see the legend of Fig. 1 for abbreviations). The number of analyzed genes ('#') per genome feature is indicated. The *p*-values of two-sided Wilcoxon rank-sum tests with continuity correction compared to the chromosome ('Chr.') are indicated above each box. Other abbreviations: 'Func. RNA (-rRNA)' (Functional RNA excluding rRNA). **d** Boxplot presenting the antisense index overgrowth in MP5 medium depending on the feature of interest. The distribution of the antisense index defined as the number of counts in antisense over total counts (in sense- and antisense- orientation) is presented depending on the genome features of interest (see the legend of Fig. 1 and panel **c** for abbreviations). The number of analyzed genes ('#') per genome feature is indicated. Please note that the antisense index was not calculated when no reads were detected, either in sense or antisense orientation, in all replicates. The *p*-values of two-sided Wilcoxon rank-sum tests with continuity correction comparing results after 24 h versus 48 h of growth for a given genome feature are indicated above each box. The *p*-values of the same tests performed to compare the antisense index of a given genome feature to the whole set of chromosomal genes after 24 or 48 h of growth are also indicated. All boxplots of this figure represent the first quartile, median and third quartile. The upper whisker extends from the hinge to the largest value no further than 1.5*the inter-quartile range (IQR, i.e. distance between the first and third quartiles) from the hinge. The lower whisker extends from the hinge to the smallest value at most 1.5*IQR of the hinge. For clarity, outliers were excluded from these graphical representations (but were taken into account for the numerical exploitation of the data).

**Compartmentalized chromosome structure in exponential phase**. We next asked the question, is a compartmentalized transcriptome correlated with a specific chromosome folding. We first performed 3C-seq on cells harvested in the exponential growth phase (Figs. 3a and 4a). The contact map displayed a main diagonal reflecting the frequency of contacts, which extends up to 1 Mb (Supplementary Fig. 5a). This diagonal contains loci acting as boundaries delimiting segments (visualized as squares, Figs. 3a and 4a) reminiscent of CIDs in other bacteria[22–24]. To define these domains precisely in *S. ambofaciens*, we computed the 'frontier index', a domain boundary indicator built from a multi-scale analysis of the contact map[40]. Briefly, two indices are computed, reflecting the intensity of the loss of contact frequencies when going downwards or upwards, respectively, to each genome position (Supplementary Fig. 5b). In this context, a boundary is defined by a significant change in both the downstream (green peaks, Fig. 3b) and upstream (orange peaks, Fig. 3b) directions (see method section for details). In the exponential phase in the YEME10 medium (non-optimal for antibiotic production, condition C6), we found 10 boundaries that defined the central region, delimited by the first and last rDNA operons, nine domains ranging in size from 240 to 700 kb (Fig. 3b). These central domains resembled regular CIDs[22,25] both in size and in the presumable nature of its formation (see below). The central boundaries are also conserved in the exponential phase in the MP5 medium, which is optimized for antibiotic production (Fig. 4a). This result indicates that the conformation of the chromosome is rather conserved in the exponential phase regardless of the media composition. Equally, the osmotic pressure present in the YEME10 medium has relatively no impact on boundary formation. Only four additional boundaries are observed in the MP5 medium (Fig. 4a). All rDNA operons coincide with the sharpest boundaries (Fig. 3b, Fig. 4a). The genes surrounding these operons, as well as those of other boundaries, are enriched in consecutive genes expressed at a very high level (Fig. 4b, Supplementary Data 3), as previously reported in other bacterial models[22,25]. Interestingly, in the exponential phase, we report a clear correlation between the level of gene conservation and the formation of boundaries (Fig. 4c). Moreover, the transcription of the genes present within boundaries tends to be oriented in the direction of continuous replication (odds ratio 1.8, *p* value $1.4 \times 10^{-8}$, two-sided Fisher's exact test for count data), with a very low antisense index (Supplementary Fig. 5c).

Genome extremities form two large terminal compartments of 1.47 and 2.50 Mb, respectively, with different structural features (Figs. 3, 4a, and Supplementary Fig. 5b). Interestingly, within the terminal arms, the low transcriptional activity in the exponential phase in the YEME10 medium correlates well with the absence of boundaries. Of note, the boundaries present in each terminal domain in MP5 medium (one of which is composed of an active prophage) do not contain highly expressed genes and do not divide the terminal compartment into two parts (Fig. 4a). To compare the dynamics of contacts within regions, we thus calculated a dispersion index, reflecting the variability of the 3C-seq signal in each region (Fig. 4d, Supplementary Fig. 5d). Despite that the plots for the probability of contacts as a function of genomic distance are similar for both the terminal and central regions (Supplementary Fig. 5e), the terminal ends present a higher dispersion index than the central compartment in both media for contacts longer than 100 kb (Fig. 4d, Supplementary Fig. 5d). This indicates that contacts between loci in the terminal regions are more variable than in the central compartment.

Thus, at the conformational level, the chromosome of exponentially growing cells of *S. ambofaciens* is partitioned into three compartments: a central compartment, actively transcribed and structured in multiple domains formed by the high-level expression of persistent genes and two large terminal compartments rich in GIs and SMBCGs and rather transcriptionally silent in which large-scale contacts are highly variable. Since in the exponential phase, the boundaries are very similar in YEME10 and MP5 media and correspond mostly to persistent CDSs (Fig. 4c) or rDNAs, their orthologous sequences may also structure the spatial organization of the chromosome in other *Streptomyces* species at a similar growth phase.

**Chromosome rearrangement during metabolic differentiation**. As shown above, during metabolic differentiation we observed that the level of transcription gradually increased within the terminal ends (Fig. 2). To determine whether transcription dynamics influence the 3D-organization of the chromosome, we performed 3C-seq on *Streptomyces* cells during metabolic differentiation in MP5 medium (48 h, condition C4) (Fig. 4e). The frontier index revealed changes in the boundaries along the primary diagonal; the boundaries identified in the exponential phase were not detected anymore after 48 h in the growth medium

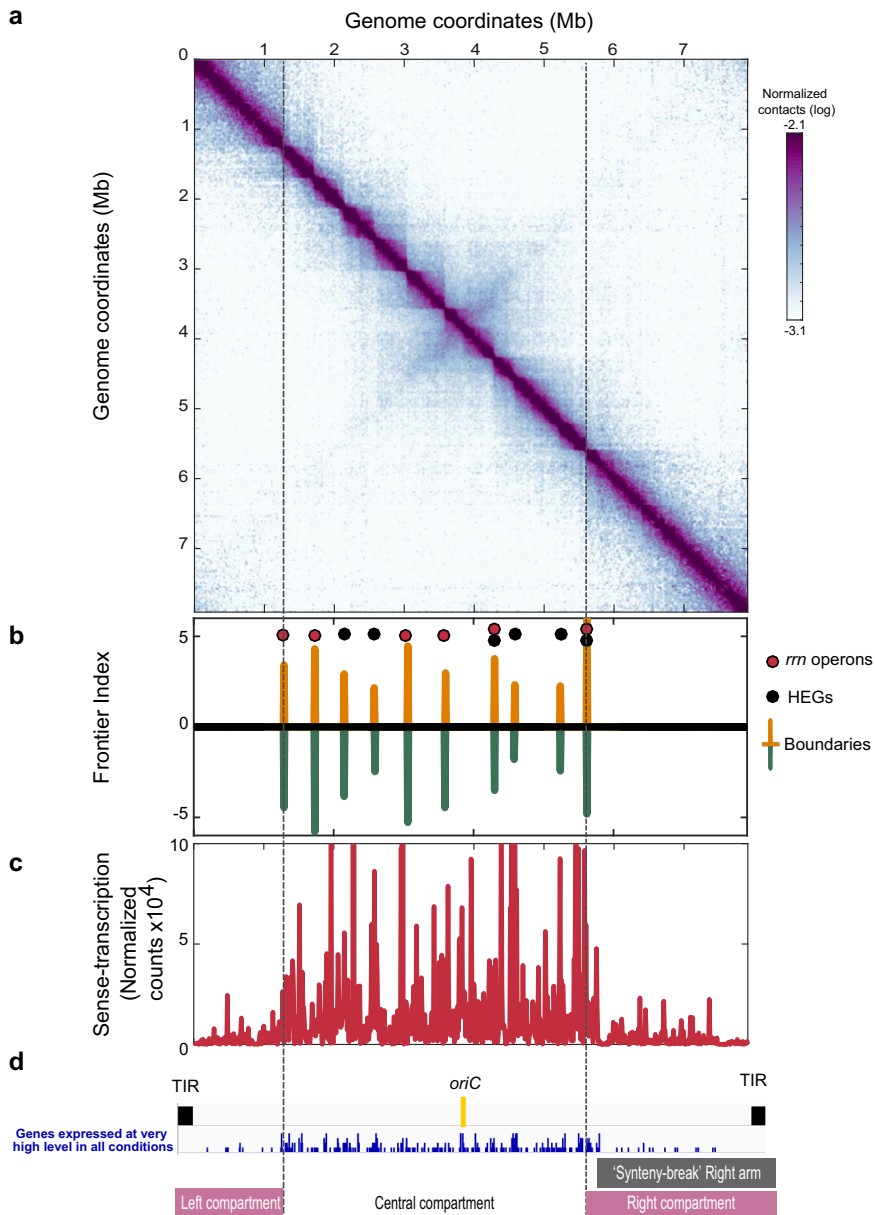

**Fig. 3 Spatial organization of _S. ambofaciens_ chromosome and transcriptome in absence of metabolic differentiation.** 3C-seq and RNA-seq were performed on _S. ambofaciens_ grown in exponential phase in YEME10 medium (24 h, C6 condition). **a** The normalized contact map was obtained from asynchronous populations. The _x_ and _y_ axes represent the genomic coordinates of the reference genome. To simplify the analyses, TIRs were removed from the reference genome. The color scale reflects the frequency of the contacts between genome loci, from white (rare contacts) to dark purple (frequent contacts). In **b**, the frontier index analysis is able to detect, for a given genome 10 kb bin, the change in the contact bias with their neighboring bins. Thus, a boundary is defined as any bin in which there is a change in the right bias of contacts towards the left bias (±2 bins, green and orange peaks, respectively). Red and black circles indicate the position of rDNAs (_rrn_) and highly expressed genes (HEGs), respectively. In **c**, the DESeq2 normalized counts measured in cells grown in the same condition were mapped on _S. ambofaciens_ chromosome and binned in 10 kb. **d** The panel highlights genomic and transcriptional features of interest. The right and left compartments were defined owing to the outer boundaries of the central compartment, which correspond to the first and last rDNA operon position (vertical lines). The 'synteny break right arm' corresponds to the beginning of the spiramycin BGC. The replicate of this experiment is presented in Supplementary Fig. 6a.

(Fig. 4e). By contrast, the appearance of a very sharp boundary within the right terminal region that divides this arm into two domains of 1.37 and 1.07 Mb, correlates very well with the increased level of transcription of the congocidine biosynthetic gene cluster (BGC, Fig. 4e, f). The congocidine BGC (29 kb) is indeed the largest SMBGC expressed at a very high level in this condition. In a replicated experiment, we observed that the expression of another SMBGC, encoding the biosynthesis of a

siderophore, could generate the formation of an additional boundary (Supplementary Fig. 6c). This illustrates some variability in the expression of SMBGCs during metabolic differentiation. Together, these results highlight the correlation between SMBGC expression and the formation of boundaries after 48 h. Accordingly, the level of conservation of genes presents within the boundaries switches from highly persistent in the exponential phase to poorly conserved during metabolic

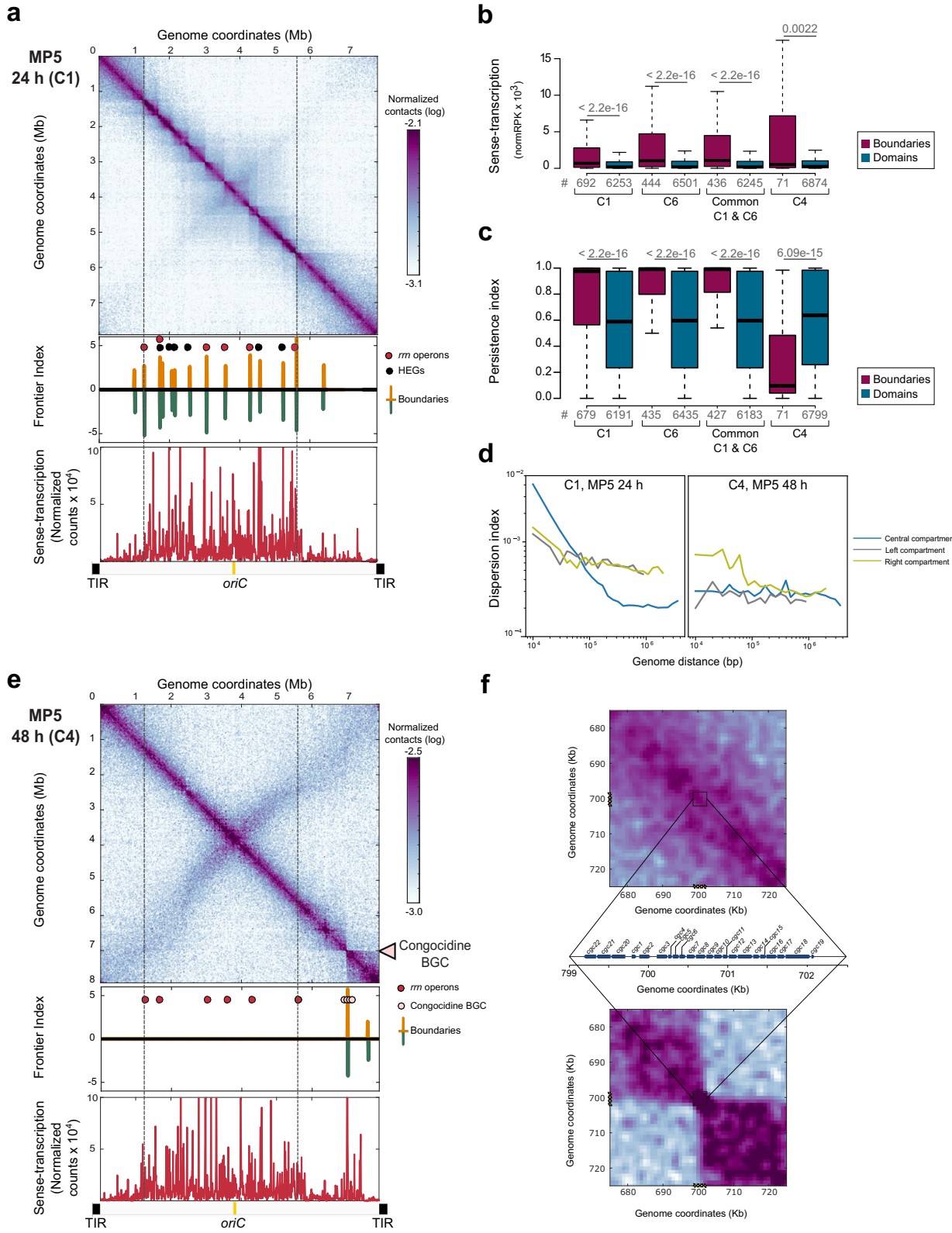

differentiation (Fig. 4c). In addition, a second boundary detected within the right compartment is located in the stambomycin BGC that is not or is poorly expressed in this condition. Interestingly, accompanying the formation of boundaries and the increase of transcription in the terminal compartments, the variability in the long-range contacts within the terminal compartments is comparable to that of the central compartment (Fig. 4d).

To gain further insights into the link between expression and local rearrangement of the chromosome induced during metabolic differentiation, we further focused on the congocidine BGC. Interestingly, congocidine (also named 'netropsin') is a pyrrolamide binding to the minor groove of the DNA double-helix in a sequence-specific manner (interaction at four or more consecutive A–T pairs)[41]. We took advantage of the fact that S.

**Fig. 4 Chromosome rearrangement during metabolic differentiation. a** 3C-seq and RNA-seq performed after 24 h in MP5 growth medium (C1 condition). Same legend as Fig. 3 except that the bacteria were grown in MP5 medium. The replicate of this experiment is presented in Supplementary Fig. 6b. **b** Level of gene expression depending on gene location within boundaries or domains. The box-plot presents the sense-transcription (DESeq2 normalized number of reading per kb, normRPK) depending on gene location in exponential phase either in MP5 (C1) or in YEME10 (C6), or after 48 h of growth in MP5 medium (C4). The expression in the C1 condition of the genes at the boundaries of domains in common between C1 and C6 conditions ('Common C1 & C6') is also presented. The p-values of two-sided Wilcoxon rank-sum tests with continuity correction (boundaries versus domains) are indicated above each box. The number of analyzed genes ('#') per condition is indicated. **c** Level of gene persistence depending on gene location within boundaries or domains. The box-plot presents the persistence index depending on gene location in exponential phase (C1: MP5; C6: YEME10; in common between both conditions: 'Common C1 & C6') or after 48 h of growth in MP5 medium (C4). The p-values of two-sided Wilcoxon rank-sum tests with continuity correction (boundaries versus domains) are indicated above each box. The number of analyzed genes ('#') per condition is also indicated. Please note that the persistence was not calculated for functional RNA encoding genes. All boxplots of this figure represent the first quartile, median and third quartile. The upper whisker extends from the hinge to the largest value no further than 1.5*the inter-quartile range (IQR, i.e. distance between the first and third quartiles) from the hinge. The lower whisker extends from the hinge to the smallest value at most 1.5*IQR of the hinge. For clarity, outliers were excluded from these graphical representations (but were taken into account for the numerical exploitation of the data). **d** Dispersion index of the left, central and right compartments plotted as a function of genomic distance. Long-range DNA contacts within the terminal compartments (>100 kb) are more variable than within the central compartment in the exponential phase. **e** 3C-seq and RNA-seq were performed after 48 h in MP5 growth medium (C4 condition). Same legend as Fig. 3 except that the bacteria were grown in MP5 medium during 48 h. The arrow indicates the position of the congocidine BGC. The replicate of this experiment is presented in Supplementary Fig. 6c. **f.** Focus on the genomic region (50 kb) encompassing the congocidine BGC contact maps (squares) after 24 h (top) or 48 h (bottom) of growth in the MP5 medium. The focus was performed on the 3C-contact maps presented in the a and e panels. The detailed genetic organization of the congocidine BGC is presented.

*ambofaciens* does not produce congocidine in the exponential phase (especially in YEME10 medium), but remains resistant to this antibiotic. The addition of exogenous congocidine (1 μg/ml) has no effect on the number and location of boundaries (Supplementary Fig. 7a). Indeed, in this condition, the transcriptome varies little with only five genes significantly induced, all belonging to the congocidine BGC (Supplementary Fig. 7b). The absence of an associated boundary highlights that the whole congocidine BGC should be expressed at a very high level (e.g. after 48 h growth in MP5) to observe a sharp boundary (Supplementary Fig. 7c).

The contact map also revealed the appearance of a secondary diagonal after 48 h of growth (Supplementary Fig. 5f), indicating an increase in the frequency of contacts between regions along the entire arms of the chromosome. This suggests that late in the cell cycle and in the absence of a central compartment segmented into multiple domains, the two chromosome arms are closer to each other (Figs. 4e and 5, Supplementary Fig. 6d). Remarkably, the second diagonal is slightly tilted when it moves away from the origin, representing an asymmetry in the contacts between loci within the terminal compartments (Fig. 4e, Supplementary Fig. 5f).

In summary, our results indicate that the central and transcriptionally active compartments in the exponential phase present no boundaries after 48 h of growth, whereas the terminal domains are locally remodeled concomitantly with the expression of SMBGCs and with a decrease in the variability of the long-range contacts. In addition, inter-arm contacts become more frequent. These results indicate that metabolic differentiation is accompanied by major rearrangement of the 3D architecture of the chromosome, both at the local and global levels (Fig. 5).

## Discussion

In this study, we demonstrated that in *Streptomyces*, compartmentalization of gene organization, transcription, and architecture are correlated. We explore the transcriptional landscape of *S. ambofaciens* ATCC 23877 during metabolic differentiation: its large genome is highly dynamic, most of the genes (≈90%) being significantly expressed in at least one condition. This situation is very similar to *Bacillus subtilis*, another Gram-positive bacteria from soil[42]. Interestingly, the SMBGCs (e.g. antibiotic clusters) are generally considered as 'cryptic' under most growth

conditions[10,43]. We observed that they are poorly expressed in the exponential phase. However, their expression is characterized by up-regulation after 48 h in the growth medium, compared to the rest of the chromosome including other genes putatively acquired by horizontal gene transfer (e.g. GIs). This result highlights that SMBGCs in *S. ambofaciens* have evolved regulatory mechanisms, such as the induction of cluster-situated transcriptional factor genes (Supplementary Fig. 3c) that efficiently and specifically regulate gene expression during metabolic differentiation, as reported in other *Streptomyces* species[4–7].

Moreover, we show that changes in gene expression dynamics are correlated with changes in the metabolome during differentiation (Supplementary Fig. 2). Indeed, the terminal regions become transcriptionally active after 48 h of growth (see also refs. [10,38]) (Fig. 2). Interestingly, the ratio, antisense- over sense-transcription, is high within the terminal compartments but decreases over the growth period, especially in the SMBGCs (Fig. 2d). Pervasive transcription has been proposed to contribute to gene regulation and genome evolution[39]. In this context, spurious transcription along the terminal arms may contribute to control gene expression during growth. This suggests that this antisense-transcription may reflect regulatory processes that remain to be explored. This is particularly interesting since some NAPs suppress antisense-transcription in *Streptomyces venezuelae*[44].

In addition, the 3C-seq analysis revealed that in the exponential phase the central compartment forms a multiple-domain structure, delineated by boundaries, whereas the terminal regions form two large compartments in which contacts are more variable at larger distances. As previously shown for bacteria with circular genomes[22–25,28], long and highly expressed genes (LHEGs encoding rRNAs, ribosomal proteins, or respiratory chain components) are found at the boundaries in the *Streptomyces* chromosome (Supplementary Data 3, Figs. 3 and 4). The positions of these boundaries are both conserved (under certain circumstances) and dynamic, since they change over the growth phase[24] (this work). Notably, most of the boundaries observed at early growth times were correlated with the presence of persistent genes (Fig. 4c). Within the central compartment, domains are a direct consequence of boundary formation. Interestingly, transcription is also a major predictor of chromosome organization into small domains throughout the Eukarya[45]. Additionally, no trivial role in gene expression or in chromosome conformation

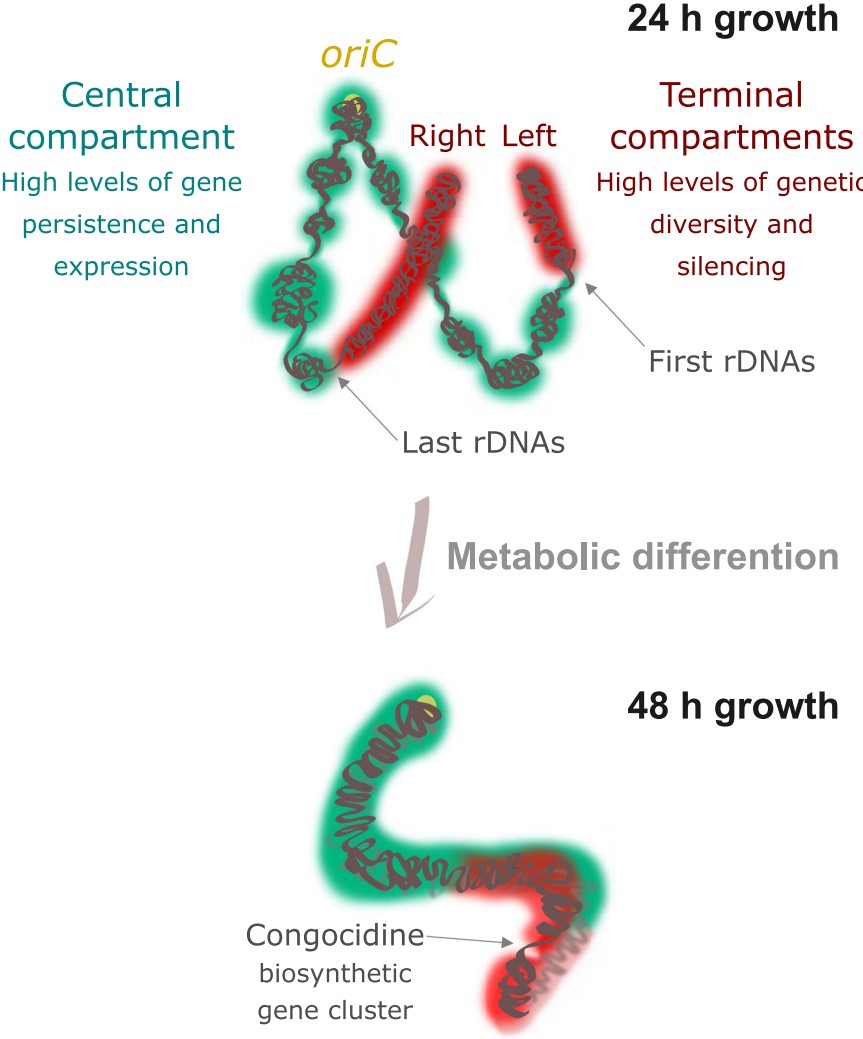

**Fig. 5 Schematic representation of genome dynamics during metabolic differentiation.** The *Streptomyces* linear chromosome is represented as green (central compartment) or red (left and right compartments). The origin of replication is indicated as a light yellow circle (*oriC*). In the exponential phase (24 h), the chromosome of *Streptomyces* is organized into one transcriptionally active (green) central compartment and two rather silent compartments (red). The transcriptionally active compartment is enriched in core genes and is segmented in multiple domains, flanked by long and highly expressed conserved genes. The frequency of contacts is not significant enough to define a genuine secondary diagonal, but the significant contacts highlight a domain encompassing the origin of replication (Supplementary Fig. 5f). The low frequency of contacts between the two arms is represented as a rather open conformation of the chromosome. After 48 h of growth, the transcription of the congocidine BCG is accompanied by a global rearrangement of the chromosome. The increase in the frequency of contacts between the two chromosomal regions around the origin is represented as a closed conformation (Supplementary Fig. 5f). Note that the inter-arms contacts observed in the stationary phase decrease toward the terminal ends (Supplementary Fig. 5f). In addition, since TIRs cannot be distinguished at the sequence level, they were neither considered in this analysis nor represented in this figure. However, they may be located close to each other[87], although this remains controversial[88]. This model is based on the 3C-map contacts of Fig. 4 and its 3D-models generated by ShRec3D software[82] (Supplementary Fig. 6d).

has been assigned to CIDs in bacteria[46]. Here we propose that boundaries are likely to act as transcriptional hubs for clustered and persistent genes[47], associated with very high levels of transcription in the exponential phase. Moreover, in the exponential phase, the first and last rDNA operons form the sharpest boundaries, recapitulating the borders that were arbitrarily used to define the central region beyond which genome synteny falls (Fig. 1). Interestingly, the percentage of the core genome present between the two most extreme rDNAs varies little (remaining close to 88.6%) compared to the relative size of the central compartment or even of the region encompassing the entire core genome (Supplementary Fig. 8). We thus propose that extreme rDNA operons constitute an evolutionary barrier that limits the occurrence of single recombination events within the central region. Interestingly, some boundaries in *Streptomyces* do not display high levels of gene expression (Supplementary Data 3). These are generally located within the terminal arms, suggesting the existence of other mechanisms that impose chromosome constraints. Silent boundaries have also been described in *B. subtilis*[23]. They correlate with Rok xenogeneic silencer-binding sites[23]. Indeed, all *Streptomyces* possess two paralogs of the xenogeneic silencer Lsr2 that belong to both the core genome and the actinobacterial signature. One of these strongly represses SMBGC expression in *S. venezuelae*[44]. Since Lsr2 is a bridging protein that can mediate DNA condensation into highly compact DNA conformation[48], its possible role in the formation of silent boundaries on very large SMBGCs (e.g. 146 kb for the stambomycin BGC) remains to be explored.

Furthermore, we show for the first time that the central and terminal compartments present different organizational features, an observation independently demonstrated for the *S. venezuelae* chromosome (see the accompanying paper, Szafran et al.). We favor the idea that the different organization of the terminal compartments could be a consequence of multiple factors: (i) the lack of constraints imposed by active and sense-transcription; (ii) the high level of anti-sense transcription; (iii) the enrichment of horizontally acquired NAPs and other transcriptional regulators that, by increasing the dynamics of DNA intramolecular interactions, ensure appropriate SMBGC repression in exponential phase. Interestingly, it was shown in *S. venezuelae* that during sporulation the HupS NAP is involved in promoting optimal contacts along the whole chromosome, but seems to be particularly important for the organization of the terminal regions (see the accompanying paper, Szafran et al.).

During metabolic differentiation, the changes in transcriptional dynamics are accompanied by a huge rearrangement of chromosome folding, switching from a rather 'open' to a 'closed' conformation, in which highly expressed SMBGC genes form new boundaries (Figs. 4e, f and 5). This may illustrate a relocation of transcriptional hubs from clustered and persistent genes (in exponential phase) to SMBGCs at the onset of metabolic differentiation. Interestingly, metabolic gene clusters have been reported to reside in dynamic chromosome 3D-domains that share common transcriptional and epigenetic states in plants[49] and probably in fungi[50]. For instance, in *Arabidopsis thaliana* BGCs change their local folding upon expression, and are associated with silent chromatin when not expressed[49]. The relocalization of SMBGCs away from a silent environment upon induction remains to be explored in bacteria.

Remarkably, the central region is no longer structured into multiple domains. The lack of boundaries in the central compartment after 48 h growth is correlated with a slight decrease in transcripts from genes that were present in boundaries in the exponential phase. Indeed most of these genes remain expressed at a high level ('CAT-3' or 'CAT_4') but below a threshold classically associated with boundaries (a string of genes with above 20,000 normalized reads per kb, Supplementary Data 3). Moreover, we cannot exclude the possibility that some of these transcripts are stable RNAs produced in exponential rather than stationary phase[51]. This loss of boundaries in the central compartment is reminiscent of the phenomena observed in eukaryotes during G1-to-mitosis transition[52] and could reflect a more general phenomenon of genome compaction during the cell cycle.

At larger scales, there is an increase in the frequency of inter-arm DNA contacts. The most likely candidate to be involved in this reorganization is the Smc–ScpAB condensin complex, which is recruited to the origin region by ParB bound to *parS*[53,54]. From the *parS* sites, Smc-ScpAB promotes inter-arm contacts by translocating to the terminal region of the chromosome, likely by loop extrusion[22,23,26,29,55–60]. *Streptomyces* spp. encode both a ParABS system[61–63] and an Smc–ScpAB complex[64]. Interestingly, during sporulation *Streptomyces* chromosomes need to be compacted and segregated to ensure correct spore formation[64,65]. Indeed, HiC studies during sporogenic development in *S. venezuelae* showed that inter-arm contacts are dependent on Smc–ScpAB (Szafran et al., see accompanying paper). The global rearrangement of the *Streptomyces* chromosome going from a rather 'open' to a 'closed' conformation seems to occur similarly during metabolic (this work) and sporogenic development (Szafran et al., see accompanying paper). Such a global rearrangement has not been previously described in wild-type bacteria. Interestingly, we observed a slight tilt in the secondary diagonal in contact analysis when it approaches to the terminal compartments (Fig. 4e, Supplementary Fig. 5f). This is consistent with a

slow-down of Smc–ScpAB activity when interacting with the transcription machinery[58,66], in agreement with the fact that transcription in the terminal compartment is higher than in the central compartment during metabolic differentiation (Figs. 2 and 4e).

Collectively, these results indicate a link between evolutionary processes, including genome-compartmentalization and the molecular mechanisms (e.g. transcription, 3D-folding) that shape the structure and function of genes and genomes in *Streptomyces*. We, therefore, hypothesize that the efficiency of the regulatory processes controlling conditional expression of SMBGCs may be an emerging property of spatial compartmentalization. We believe that this study will open new insights into setting the rules governing chromosome spatial organization, expression and stability, and the optimal design of *Streptomyces* genomes for SMBCG expression.

## Methods

**Streptomyces strains and growth conditions**. *S. ambofaciens* ATCC 23877 was grown on solid soy flour-mannitol (SFM) medium[67] at 28 °C unless otherwise indicated. The strain was grown in the following media: MP5 medium (7 g/l yeast extract, 20.9 g/l MOPS, 5 g/l NaCl, 1 g/l NaNO₃, 36 ml/l glycerol—pH 7.5)[36], or a modified version of the YEME medium[67] that we term "YEME10", containing only 10.3% sucrose (3 g/l yeast extract, 5 g/l bactotryptone, 3 g/l malt extract, 10 g/l glucose, 103 g/l sucrose; pH 7.0–7.2). Twenty million spores of *S. ambofaciens* ATCC 23877 were inoculated in 100 ml of media before growth at 30 °C in a shaking agitator (220 rpm, INFORS HT multitron standard). When appropriated 1 μg/ml congocidine (Sigma) was added in the medium.

**Bioassays**. For bioassays from liquid cultures, 50 μl of supernatants (filtered using Whatman PVDF Mini-UniPrep syringeless filters, 0.2 μm) were spotted on to agar medium using cylinder (diameter 0.5 cm) and allowed to dry until complete penetration into a plate containing 50 ml of DIFCO antibiotic medium 5 (DIFCO 227710). Thereafter 7 ml of SNA medium (2 g/l agar, 8 g/l nutrient broth MP Biomedicals Cat#1007917) containing *Micrococcus luteus* (final OD₆₀₀ nm 0.04) were overlaid on the plate and incubated at 37 °C. The growth inhibition area was measured 24 h later.

**HPLC analyses**. After 24 and 48 h of growth, supernatants were filtered through Mini-Uniprep syringeless filter devices (0.2 μm, Whatman) and analyzed on an Atlantis Dc18 100 A (5 μm 4.6 mm × 150 mm) column (temperature 31 °C) using a Dionex Ultimate 3000 HPLC instrument (Thermo Scientific). Samples were 5-fold diluted in a solution of 20 μM cyclo(Trp-Trp). This latter was used for sample normalization since its retention time of 24.8 min is outside the window at which the detectable metabolite peaks produced by *S. ambofaciens* emerge. Samples were separated using the following method: 0 to 7 min, isocratic 0.1% HCOOH in water (solvent A)/0.1% HCOOH in CH₃CN (solvent B) (95:5) at 1 ml min⁻¹, 7–30 min, 5–60 % solvent B, 30–35 min, 60–100% solvent B, 35–45 min, 100% solvent B and 50–60 min, 5% solvent B. Metabolite production was detected at 297 nm.

**GI identification**. We designed the Synteruptor program (http://bim.i2bc.paris-saclay.fr/synteruptor/, v1.1)[68] to compare the sequences of chromosomes of species close enough to have synteny blocks and to identify the GIs existing in each respective chromosome. We define synteny breaks as genomic regions between two consecutive synteny blocks when comparing two genomes, with the two blocks having the same direction. The genome of *S. ambofaciens* ATCC 23877 was compared to the chromosome of 7 closely related strains: *S. ambofaciens* DSM40697, *S. coelicolor* A3(2), *S. lividans* 1326, *S. lividans* TK24, *S. mutabilis*, *Streptomyces* sp. FXJ7.023, *Streptomyces* spp. M1013. To define a region harboring GIs, we used a threshold of 15 CDS as the minimal number of CDSs within the synteny break, in at least one of the strains used for the pairwise comparison. Of note, a GI can therefore correspond to 15 CDS in one strain but fewer CDS in the other one. However, when the corresponding position within the *S. ambofaciens* genome contained <2 CDS or only tRNAs, compared to at least 15 CDSs in the chromosome of the compared species, it was considered as an insertion point (but not a GI) within *S. ambofaciens*. When the GIs were identified during several pairwise comparisons of *S. ambofaciens* ATCC 23877 and/or overlapping, they were fused and considered thereafter as a single GI. The complete list of GIs identified in the *S. ambofaciens* genome is presented in Supplementary Fig. 1 and Supplementary Data 1.

**SMBGC identification**. We used antiSMASH5.1.0[35] to identify putative SMBGCs. Thirty SMBGCs have thus been identified in the *S. ambofaciens* genome, the one encoding kinamycin biosynthesis being duplicated since this SMBGC is located within the TIRs. The definition of cluster boundaries has been manually defined on

the basis of literature data for characterized SMBGCs (Supplementary Data 2). Of note, SMBGC genes are ≈1.5 times larger than the average, and only 13 of them (from 4 SMBGCs, namely 'CL4_Indole', 'CL10_Furan', 'CL11_NRPS', 'CL20_Hopenoid') belong to the core genome.

**Identification of NAPSFs.** In this study, we considered NAPs and chromosome structural factors *sensu lato* by including orthologues of classical NAPs and structural factors (HU, sIHF, Lsr2, Lrp/AsnC, SMC, Dps, CbpA, DnaA, or IciA family proteins)[69] and/or proteins associated with *S. coelicolor* chromatin[70]. This list is available in Supplementary Data 3.

**Definition of indices for genome conservation analyses.** We selected 125 *Streptomyces* genomes from the NCBI database representative of the *Streptomyces* genus by keeping only complete genomes of distinct species. When genomes share an average nucleotide identity (ANIb) value greater than or equal to 96%, they are considered as members of the same species[30] (Supplementary Table 1). We made one exception by keeping two strains of *S. ambofaciens* (ATCC 23877 and DSM 40697). Orthologous genes were identified by BLASTp best hits[71] (BLASTP 2.11.0+) with at least 40% of identity, 70% coverage and an *E*-value lower than 1e$^{-10}$ as previously described[30]. The gene persistence index was calculated as $N_{orth}/N$, where $N_{orth}$ is the number of genomes carrying a given orthologue and $N$ is the number of genomes searched[31]. For gene order conservation (GOC) analysis, pairwise comparisons were achieved using a sliding window (8 CDSs, step of 1 CDS) and comparing a reference strain (*S. ambofaciens* ATCC23877) to another. The GOC index is the number of contiguous orthologs in both chromosomes over the number of orthologs in the window.

**Transcriptome analysis.** For RNA-seq analysis performed in liquid cultures, ≈2 × 10$^7$ spores of *S. ambofaciens* ATCC 23,877 were inoculated in 100 ml of liquid medium. Thereafter, 25 ml (for samples harvested after 24 h growth) or 10 ml of cultures (for other time points) were added to an equal volume of cold ethanol.

Cells were then harvested by centrifugation for 15 min at 4000 × *g* at 4 °C, and stored at −20 °C. Pellets were washed with 1 ml of DEPC water, centrifuged for 5 min at 16,000 × *g* at 4 °C and homogenized with glass beads (≤106 µm; G4649, SIGMA) in 350 µl of lysis buffer (RNeasy Plus Mini Kit, QIAGEN) supplemented with 10 µl/ml β-mercaptoethanol. Samples were processed 3 times for 45 s each in FastPrep-24TM 5 G instrument (MP Biomedicals) at setting 6 with 1 min cooling between the stages. After centrifugation for 10 min at 16,000 × *g* at 4 °C, total RNAs were isolated from the supernatants using an RNeasy Plus Mini Kit (QIAGEN) and gDNA Eliminator columns, following the manufacturer's recommendations. To remove genomic DNA, RNA samples were incubated for 30 min at 37 °C with 20 U of RNase-free DNase I (Roche) in a final reaction volume of 30 µl. RNAs were then cleaned up using the RNeasy Mini Kit (QIAGEN), following the manufacturer's recommendations. The absence of DNA in the preparations was checked by PCR on an aliquot. RNA samples were quantified using QubitTM RNA HS Assay kit (Thermo Fischer Scientific), following the manufacturer's recommendations.

Total RNA quality was assessed in an Agilent Bioanalyzer 2100, using RNA 6000 pico kit (Agilent Technologies). Five hundred ng of total RNAs were treated with DNAse (Baseline Zero DNAse, Epicentre) prior to ribosomal RNA depletion using the RiboZero bacteria magnetic kit from Illumina according to the manufacturer's recommendations. After the Ribo-Zero step, the samples were checked in the Agilent Bioanalyzer for complete rRNA depletion. Directional RNA-seq libraries were constructed using the Illumina ScriptSeq kit V2 (discontinued) for samples corresponding to C5 conditions and Illumina Stranded library preparation kit for all other samples, according to the manufacturer's recommendations.

Libraries were pooled in equimolar proportions and sequenced (Paired-end 2 × 43 bp) with an Illumina NextSeq500 instrument, using a NextSeq 500 High Output 75 cycles kit. Demultiplexing was done (bcl2fastq2 V2.2.18.12) and adapters (adapter_3p_R1: AGATCGGAAGAGCACACGTCTGAACT; adapter_3p_R2: AGATCGGAAGAGCGTCGTGTAGGGA) were trimmed with Cutadapt1.15, only reads longer than 10 bp were kept.

STAR software[72] (v2.5.4) was used for mapping RNA-seq to the reference genome (genome-build-accession NCBI_Assembly: GCF_001267885.1) containing only one terminal inverted repeat (TIR). This avoids any biases with multiple mapping within the duplicated extremities of the genome (since the two TIR sequences are indistinguishable). We used the featureCounts program[73] (v2.0.1) to quantify reads (in sense- and antisense-orientation) generated from RNA-sequencing technology in terms of "Gene" feature characteristics of *S. ambofaciens* ATCC 23877 annotation (GCF_001267885.1_ASM126788v1_genomic.gff—released on 06/15/2020).

**Bioinformatic analysis of RNA-seq count data.** SARTools (Statistical Analysis of RNA-Seq data Tools, v1.6.3) DESeq2-based R pipeline[74] was used for systematic quality controls and the detection of differentially expressed genes. PCA and sample clustering used homoscedastic data transformed with Variance Stabilizing Transformation (VST). For the differential analysis, Benjamini and Hochberg's method was used with a threshold of statistical significance adjusted to 0.05. The reference condition was C1 (medium "MP5_24h"). All parameters of SARTools

that have default values were kept unchanged. Genes with null read counts in all samples were not taken into account for the analysis with DESeq2. Genes corresponding to the second TIR (not used for the mapping) or the rRNAs (RiboZero treatment) were excluded from further analysis. The statistical report of this RNA-seq analysis is presented in Supplementary Data 4. Supplementary Data 3 presents the DESeq2 normalized counts per kb for each gene in each growth condition. To ensure that antisense-transcription did not affect the normalization of sense-transcription data, the latter were normalized as follows: the percentage of transcription in antisense orientation of each gene was determined with the raw data, then the normalized antisense-transcription was evaluated by applying this percentage to the normalized sense-transcript counts. To determine the category of gene expression the normalized counts obtained by the SARTools DESeq2-based pipeline were again normalized with respect to gene size (number of DESeq2 normalized reads/gene size × 1000 bp, normRPK). This allowed us to compare directly the relative levels of gene expression of individual genes. Data were analyzed with R software[75] and the Integrative Genomics Viewer (IGV, v2.8.0) tool was used to simultaneously visualize RNAseq data and genomic annotations[76].

**Multidimensional analyses of the data.** We used the FactoShiny R package[77] (v2.4) to perform clustering, principal component analyses, and correspondence analyses. This overlay factor map presented in Supplementary Fig. 4c results from the correspondence analysis performed using a contingency table indicating the number of genes in each category of expression level depending on the genome features of interest. 'Max', 'Mean', and 'Min' refers to the maximal, mean, and minimal expression levels in all studied conditions, from the lowest category ('0') to the highest ('4'). These categories were defined by considering the distribution parameters of the normalized number of reads (Supplementary Fig. 4a). These gene expression categories as well as the number of genes switched ON ('Switch', meaning that the expression level switches from CAT_0 in at least one condition to CAT_3 or more in another condition), or presenting a very low ('AS < 0.05') or high ('AS > 0.5') level of antisense index were used to build the map, and projected on the plan.

**Chromosome conformation capture (3C).** Spores of *S. ambofaciens* ATCC 23877 (≈4 × 10$^7$) were inoculated in 200 ml of MP5 or YEME10 liquid medium. In these media, bacterial growth was monitored by opacimetry. At the indicated time point, cells were harvested from 100 ml samples of culture, adjusted to an OD$_{600\,nm}$ of about 0.15 (with the appropriate fresh medium), and fixed by adding formaldehyde solution (3% final concentration). Cells were then incubated under moderate agitation for 30 min at RT and 30 min more at 4 °C. Glycine (250 mM final concentration) was added, and the bacteria incubated 30 min at 4 °C with moderate agitation. Cells were then harvested by centrifugation for 10 min at 4000 × *g* at 4 °C. The cells were gently suspended in 50 ml of PBS 1× and then again harvested by centrifugation for 10 min at 4000 × *g* at 4 °C. This washing step was repeated once before suspending the cells in 1 ml of PBS before final harvesting by 10 min at 4000 × *g* at 4 °C. The dry pellets were stored at −80 °C until use.

Frozen pellets of exponentially grown cells were thawed, suspended in 600 µl Tris 10 mM EDTA 0.5 mM (TE) (pH 8) with 4 µl of lysozyme (35 U/µl; Tebu Bio) and incubated at RT for 20 min. For the samples collected after 48 h, the pellets were thawed, suspended in TE (pH 8) with 4 µl of lysozyme (35 U/µl; Tebu Bio) for 45 min and then homogenized with a Bioruptor sonication device (3 cycles of 30 s, with a pause of 30 s pause for each). Then, for both 24 h and 48 h samples, SDS was added to the mix (final concentration 0.5%) of cells and incubated for 10 min at RT. Five hundred µl of lysed cells were transferred to a tube containing 4.5 ml of digestion mix (1× NEB 3 buffer, 1% triton X-100) and 100 µl of the lysed cells were transferred to a tube containing 0.9 ml of digestion mix. Eight hundred units of *Sal*I were added to the 5 ml digestion mix. Both tubes were then incubated for 2 h at 37 °C and 250 units of *Sal*I were added to the 5 ml tube and further incubated, 2 h at 37 °C. To stop the digestion reaction, 4 ml of the digestion mix were immediately centrifuged for 20 min at 20,000 × *g*, and pellets were suspended in 4 ml of sterile water. The digested DNA (4 ml in total) was split into 4 aliquots and diluted in 8 ml ligation buffer (1× ligation buffer NEB 3 (without ATP), 1 mM ATP, 0.1 mg/ml BSA, 125 Units of T4 DNA ligase 5 U/µl). Ligation was performed at 16 °C for 4 h, followed by incubation overnight at 65 °C with 100 µl of proteinase K (20 mg/ml) and 100 µl EDTA 500 mM. DNA was then precipitated with 1/10 volume of 3 M Na-acetate (pH 5.2) and two volumes of isopropanol. After one hour at −80 °C, DNA was pelleted and suspended in 500 µl 1× TE buffer. The remaining 1 ml digestion mix with or without *Sal*I were directly incubated with 100 µl of proteinase K (20 mg/ml) overnight at 65 °C. Finally, all the tubes were transferred into 2 ml centrifuge tubes (8 tubes), extracted once with 400 µl phenol-chloroform pH 8.0, precipitated, washed with 1 ml cold ethanol 70% and diluted in 30 µl 1× TE buffer in the presence of RNAse A (1 µg/ml). Tubes containing the ligated DNA (3C libraries) were pooled. The efficiency of the 3C preparation was assayed by running aliquots of the 3C-libraries, the digested DNA or the non-digested DNA on 1% agarose gel. Finally, the 3C libraries were quantified on the gel using QuantityOne software (BioRad).

**Processing of libraries for Illumina sequencing.** Approximately 5 µg of a 3C library was suspended in water (final volume 130 µl) and sheared using a Covaris

S220 instrument (Duty cycle 5, Intensity 5, cycles/burst 200, time 60 s for 4 cycles). The DNA was purified using a Qiaquick® PCR purification kit, DNA ends were prepared for adapter ligation following standard protocols[78]. Custom-made adapters[23] were ligated overnight at 4 °C. Ligase was inactivated by incubating the tubes at 65 °C for 20 min. To purify DNA fragments ranging in size from 400 to 900 pb, a PippinPrep apparatus (SAGE Science) was used. For each library, one PCR reaction of 12 cycles was performed using 2–3 μl of 3C library, 0.2 μM Illumina primers PE1.0 and PE2.0, and 1 unit of Taq Phusion (Finnzymes). The PCR product was purified on Qiagen MinElute columns and dimers of primers were removed from the 3C library by using AMPure XP beads following the manufacturer's protocol (Beckman Coulter). Finally, libraries were subjected to 75 bp paired-end sequencing in an Illumina sequencer (NextSeq500).

**Processing of sequencing data**. PCR duplicates from each 3C library sequence dataset were discarded using the 6 Ns of custom-made adapters[23]. Reads were aligned independently using Bowtie 2 (v2.2.3) in very sensitive mode[79]. Only reads with mapping quality >30 were kept to establish contact maps.

**Generation and analyses of contact maps**. Contact maps were built as described previously[80]. Briefly, each read was assigned to a restriction fragment. Non-informative events such as self-circularized restriction fragments, or uncut co-linear restriction fragments were discarded[81]. The chromosome of *S. ambofaciens* devoid of the TIRs was divided into 10 kb bins and the frequencies of contacts between genomic loci for each bin were assigned. Contact frequencies were visualized as heatmaps. Raw contact maps used for comparative analyses were built using ≈2 million valid reads. Raw contact maps were normalized with the sequential component normalization procedure (SCN)[81]. To facilitate visualization, contact matrices are visualized as log matrices. First, we applied to the SCN matrices the log10 and then a Gaussian filter ($H = 1$) to smooth the image. The contact score for a pair of bins that due to mapping was identified as an outlier ($z$-score > contact score) was replaced with the median value of the contact matrix. We built our model (Fig. 5) based on the 3D representations of the contact map using ShRec3D[82] already used in other bacteria[23,24] (Supplementary Fig. 6D). The resulting model is a 3D representation of the 2D contact map and does not reconstruct 3D structures based on a polymer model neither represent the exact nucleoid structure found in an individual bacterium but represents a convenient visualization tool of contact frequencies over a population of cells. To quantify the extent of the primary and secondary diagonals (Supplementary Fig. 5F), we applied to our data a method based on the filtering of significant 3C contacts and developed by Wang et al.[55,83].

**Frontier index determination**. To analyze the domain organization of *S. ambofaciens*, we used the frontier index (FI) method that quantifies the involvement of each bin in the frontier of any domain, i.e. at any scale—see technical details in ref. [40]. Briefly, this method consists, first, in computing derivatives of the contact maps. To this end, we used normalized contact maps where the average contact frequency, $P(s)$, was substracted to reduce noise coming from the natural overall decrease of contact frequencies as the genomic distance ($s$) between loci increase. Here, we also added pseudocounts (equal to $m/20$ where $m$ is the maximal value of the contact frequencies) such that all values of the contact maps are strictly positive and we considered the logarithm of the resulting maps in order to mitigate the strongest variations close to the diagonal. We then considered separately the derivatives of these maps along the vertical and horizontal axes, whose large positive values (in the upper part of the maps) are, respectively, associated with upstream and downstream frontiers of domains, respectively. After setting negative values to zero (to reduce noise even more), for each bin, the corresponding upstream and downstream FIs were defined as the sum over all other bins at a distance below 600 kb of the resulting signals. We then computed the profile of upstream and downstream FIs. We considered peaks that are located above the median of the peak values plus two times the standard deviation (estimated by 1.48 times the median absolute deviation to mitigate the impact of outliers). These statistics were computed separately for the left terminal arm, the central region, and the right terminal arm as these compartments have distinct statistical 3C-seq features. Altogether, this led us to a list of significant upstream (orange) and downstream (green) peaks, respectively.

A boundary was then allocated to any bin (as well as to pairs of consecutive bins to cope with the uncertainty of peak positions) for which both the upstream and downstream peaks were significant. To analyze the genetic environment associated with a boundary, we considered the environment around the boundary identified by the frontier index (±2 bins). Note that due to the intrinsically noisy nature of 3C-seq data, we detected bins with only an upstream or a downstream significant peak (Supplementary Fig. 6E). These bins with an 'orphan' peak were not associated with a boundary, and hence, were removed from the analysis.

**The dispersion index**. The dispersion index of a signal reflects the range of variations relative to the mean value of the signal. Here, we compute it for the frequency of contacts. More precisely, it is defined as: $I(s) = \mathrm{var}(s)/P(s)$, where $P(s)$ and $\mathrm{var}(s)$ are, respectively, the mean value and variance of the frequency of contacts between bins separated by the genomic distance. These quantities are then computed separately in each compartment of the genome. The larger the dispersion index is, the more variable are the contact frequencies within a compartment.

**Statistical procedure**. Data were analyzed with R software[75]. For the RNA-seq analyses, three independent experiments (performed on different days) were carried out for each studied condition, except for C3, C9, and C10 that were performed in duplicate and C5 in quadruple. The statistical significance of RNA-seq analysis was assessed using the SARTools DESeq2-based pipeline[74]. For the 3C-seq analyses, two independent experiments (performed on different days) were carried out for each condition. To quantify gene regionalization, we defined the left and right terminal arms using as limits the first (genome position: 1,469,670–1,474,823 bp) and last (genome position: 5,802,471–5,807,629 bp) rDNA operons, respectively. The statistical significance was assessed by means of two-sided Fisher's exact test for count data, which is appropriated to the analysis of contingency tables.

**Reporting summary**. Further information on research design is available in the Nature Research Reporting Summary linked to this article.

## Data availability
The RNA-seq and 3C-seq data generated during in this study have been deposited in the NCBI Gene Expression Omnibus (GEO, https://www.ncbi.nlm.nih.gov/geo/) under the accession code GSE162865. The list of the core genome CDSs and the persistence index values are available in Supplementary Data 3. Source data are provided with this paper.

## Code availability
The scripts used for data analyses are available on the following Github links: RNA-seq analyses (https://github.com/PF2-pasteur-fr/SARTools)[74], 3C-seq contact-map analyses (https://github.com/koszullab/E_coli_analysis), 3C-seq frontier index analyses (https://github.com/VickyTche/Frontier_Index_Streptomyces.git; https://osf.io/a23de/; https://doi.org/10.17605/OSF.IO/A23DE[84]), persistence and core genome analyses (https://github.com/jnlorenzi/pipeline-core-and-persistence; https://doi.org/10.5281/zenodo.506721[85]), Synteruptor software associated code (https://github.com/jnlorenzi/synteruptor; https://doi.org/10.5281/zenodo.5080081[86]).

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

## Acknowledgements

We acknowledge the High-throughput sequencing facility of I2BC for its sequencing and bioinformatics expertise. We thank Barry Holland and Christophe Possoz for careful reading of the manuscript and the members of F.B. and S.L. for fruitful discussions and advice. We thank the I2BC for the attribution of a starting grant.

## Author contributions

Supervision and design of the experiments: V.S.L., S.B.-M.; Investigation: V.S.L., S.B.-M., H.L., C.S., S.N., Y.J., K.G.; Synteruptor software development: J.-N.L., O.L., B.A., P.L., A.T., S.L., J.-L.P.; Bioinformatic analyses and script development: V.S.L., S.B.-M., J.-N.L., T.P., N.V., I.J.; Writing—Original draft: V.S.L., S.B.-M., F.B., J.-L.P., N.V., I.J.; Writing—Reviewing and Editing: all authors; Funding acquisition: J.-L.P., S.L., P.L., O.L., B.A., S.B.-M., V.S.L., F.B.

## Competing interests

The authors declare no competing interests.
