## [Peer Review File · Nature Communications]

Dynamics of the compartmentalized *Streptomyces* chromosome during metabolic differentiationREVIEWER COMMENTS

Reviewer #1 (Remarks to the Author):

In this study from the Bury-Moné Lab, Liroy et al investigated *Streptomyces* genome organization during metabolic differentiation. They analyzed the genome sequences of 125 species of *Streptomyces* and found that the center of the chromosome are enriched with core genes and the terminus region are enriched with metabolic genes. They performed transcriptome analyses and found that the metabolic genes are repressed during exponential growth and expressed during differentiation. Finally, they performed genome-wide chromosome conformation capture (3C-seq) experiments to investigate chromosome organization during exponential growth and differentiation.

The mechanism of chromosome organization and remodeling during differentiation remains an important and fundamental problem. It was previously reported that *Streptomyces* genomes are compartmentalized into central regions and terminal regions (Choulet 2006), and genes in these regions are differentially expressed in various growth conditions (Karoonthaisiri 2005, Jeong, 2016). Using 3C-seq, the authors found that the chromosome forms CIDs bounded by highly transcribed genes during exponential growth and the CIDs boundaries disappeared during development (which was essentially prolonged growth/stationary phase). This correlation of gene expression and CID boundaries were previously described in multiple bacterial species (Le 2013; Le 2016; Marbouty 2015; Liroy 2018; Bohm 2020, and more). Therefore, this manuscript confirms and extends previous findings but do not provide many new insights. It is interesting that the strength and extense of inter-arm contact increased during development compared to exponential phase. If they focused on the mechanism and biological relevance of this the change, the authors could have increased the impact of the work. There are some very straight forward experiments to address these. For example, are ParB/parS Smc-ScpAB required for the inter-arm contacts? Do cells have a higher level of these proteins during development? Is the increased inter-arm contact a consequence of the reduced transcription at stationary phase, i.e. if transcription is inhibited during exponential phase, would inter-arm contact increase? Does inter-arm contact have an impact on development, i.e. does losing inter-arm contact (in ParB/Smc mutants) have a negative effect on the development? Without these answers, this manuscript does not add much to the canon.

Minor points:

- 1) "Remodeling" indicates an active mechanism. However, the change of chromosome organization observed here could be passive, i.e. a byproduct of changed transcription. Without identifying a factor that remodels the chromosome, the authors should tune this down.
- 2) Changing from "open" to "closed" conformation is an overstatement. In exponential phase (Fig 3A), there is a faint off-diagonal, so the chromosome is not really "open", but is weakly closed.
- 3) Figure 4F is not supported by the data. In exponential phase, there is some inter-arm contact in the data that is missing in the model. After 48 h of growth, only the regions near the origins are weakly interacting and these interactions do not extend to the entire genome. The authors should make the inter-arm contact weaker and shorter, not the strong alignment of the entire arms.

Reviewer #2 (Remarks to the Author):

Streptomyces are prolific antibiotic producers that are of great importance for society. About two-thirds of all antibiotics are derived from these bacteria. Unlike most other bacteria, streptomycetes have linear chromosomes, and many of gene clusters required for synthesis of these important metabolites are located on the so-called terminal ends of the chromosome. Despite decades of work, very little is known about the organisation of the linear chromosomes in these bacteria during growth. In this manuscript, the authors perform a very thorough investigation in the dynamics and organisation of the chromosome during differentiation. They use

3C-seq to demonstrate that the chromosome ends are relatively silent during primary growth, but become active during secondary metabolism whereby the chromosome also adopts a different conformation. In the latter state, the chromosome ends are more frequently in proximity of one another than during primary growth. The authors also show that the central part of the chromosome is organised in regions that are transcriptionally active and which are defined by so-called boundaries (such boundaries are also evident in "older cultures" when more active transcription is detected here). The manuscript is well written and presented. The work also provides important new insights that are interesting for a broad readership. It forms a foundational story that can lead to many interesting follow-up stories.

I have some suggestions to improve the manuscript.

1. Introduction/Results: The authors have chosen to do their work in *Streptomyces ambofaciens*. Although well-studied, this is not one of the canonical models in the field. Although this is of course not a problem, it would be good to describe in the introduction/beginning results why this model was chosen and not for instance *S. venezuelae* or *S. coelicolor*, which are more commonly used in the field.
2. The arms of both chromosomes are quite different in length. Can the authors perhaps further explain how the disappearance of synteny (as described in Fig. 1, lines 104-105) relates to the different arms?
3. The use of Extended Data Figures and Supplementary Figures is confusing. I expect that this will be dealt with in the final version?
4. The authors seem to have used a modified version of YEME as mentioned in lines 388-390 (please indicate this clearly, as normal YEME contains 34% sucrose). Also, this medium is difficult to consider as a really biological relevant medium. Specifically, the medium imposes dramatic osmotic stress on the cells. How does this influence the organisation of the chromosome? Perhaps the authors can comment a bit on this in the discussion?
5. I am curious if the authors think whether the change in chromosome conformation also relates to the genomic instability that is often witnessed in streptomycetes. The reason I mention this is because the authors seem to suggest that the high level of transcription in the central part of the chromosome, flanked by the boundaries is perhaps less prone to recombination (lines 318-320). Does this also mean that the recombination events (and genomic instability in the chromosomal arms) occurs during early growth (because those arms are not active then)?

Minor comments

1. Line 29. Change to "...chromosome structure of *Streptomyces ambofaciens* correlates with..."
2. Line 46. The authors use e.g. However, what other forms of differentiation are meant? I think the authors mean i.e.?
3. Lines 60-64. The authors could include some references to work from other labs that have demonstrated genomic rearrangements (although admittedly the cited lab is leading in this respect).
4. Line 83-84 Change to: "...the numerous available *Streptomyces*"
5. Line 85. Omit "new" (it may be new for *Streptomyces*, but it is commonly used in other systems)
6. Line 115. Change "shorten" to "referred to as"
7. Line 120. Change to "terminal" (not terminals)
8. Line 138. remove "by *S. ambofaciens*"
9. Line 395. What filters were used?
10. Line 428. "owing to its location within the TIRs". This is not clear. Please rephrase
11. Line 434. Change to "Identification of NAPSFs"
12. Line 448. $1e-10$ 31?
13. Line 452-458. Split into 2-3 sentences for clarity.
14. Line 802. Same legend than Figure 3?
15. Line 829. Change "under" to "in"
16. Line 830. Omit "although"
17. Line 841. I guess ATB is antibiotic?
18. Fig. 1. "Diversity" is without capital
19. Ext. Data Fig. 1. Why did the authors not include a 72h YEME sample? It seems that the halo

size is quite a bit bigger than at 48 h in YEME.

Reviewer #3 (Remarks to the Author):

In the manuscript under review, Lioy et al report an investigation into chromosome structure in *Streptomyces*. The authors focus their investigations on metabolic differentiation and highlight biosynthetic gene clusters. The organisation of genes related to specialised metabolism into clusters, groups of neighbouring functionally related genes, is a widespread phenomenon throughout both pro- and eukaryotic genomes. How chromosome folding in the three-dimensional cellular spaces relates to this organisational form is a largely untapped area of science. As such, the work presented here, represents a major milestone in bacterial genomics and *Streptomyces* biology.

The first results section describes sequence compartmentalisation of the *S. ambofaciens* genome. These are largely standard analyses and confirm the expected pattern of gene distribution in *Streptomyces* genomes.

The transcriptional analyses are powerful and clear. They underscore previous findings in *Streptomyces* on the transcriptional compartmentalisation of the chromosome.

The 3Cseq analysis enters new territory in *Streptomyces* genomics. The authors show data for three different conditions (2x bacterial growth phase and 1x stationary phase) and correlate their Hi-C maps with the transcriptional analyses. The presented data reveals a chromosome compartmentalisation largely reflective of the sequence and transcriptional organisation. The central chromosomal region (transcriptionally active) is separated from the two chromosomal ends (transcriptionally inactive) during the growth phase. In addition, distinct domains are formed throughout the central region. Boundaries to these domains are enriched in transcriptionally highly active operons and highly conserved genes. During stationary phase, a new pattern of chromosome structure can be observed. The central chromosome region is not occupying a distinct domain anymore and instead novel boundary regions can be observed at two biosynthetic gene clusters. Furthermore, increased contacts between the chromosomal ends are detectable according to the 3Cseq data presented.

Overall, an array of very solid data is presented in this manuscript. Data on genomics, transcriptomics and 3D chromosome structure are well-aligned to each other and the authors have been able to assemble a coherent story.

However, I am struggling to connect metabolic differentiation with the presented data. There is no metabolite data shown and strong activation of a single BGC appears somewhat limited evidence for metabolic differentiation. The authors claim activation of 20 % of putative SMBGC genes during stationary phase but it remains unclear to which extent this relates to 'real' metabolic changes. To me, chromosomal re-arrangements are clearly correlated to growth phase but, with the current data provided, only to limited extent to metabolic output.

Furthermore, I see limited support presented here for a better understanding of BGC regulation. The authors provide two examples for an overlap between BGC location and boundary formation. Given that there seemingly are around 20 BGCs in the investigated species, the finding of just two boundary overlaps appears limited. One of the two examples shows very high expression and in the other example rather low expression. As such, it remains unclear how BGCs influence domain formation. The role of boundary formation in cluster regulation remains unclear as well (see specific question below).

As a pioneering study for *Streptomyces* genetics, the manuscript would be significantly strengthened if the authors could provide any type of confirmatory experiments to their 3Cseq chromosome architecture data. DNA-FISH is the gold standard in the community and appears feasible to present evidence for the different chromosomal condensation states in the tested conditions. If FISH prove to difficult to be carried out, it maybe possible to present locus specific 3C data. This would appear particularly useful for the Congocidin BGC.

Specific comments.

1. Fig 2a is unclear. Does each line represent a gene?
2. L 147 – expressed and/or regulated – unclear.
3. L 163-176 – antisense transcription – could the authors please expand. It is unclear how to align the positive correlation of antisense to sense transcription with the highest antisense index at lowly expressed genes. The authors suggest a role a gene regulation based on the presented data but avoid to outline a brief mechanistic model.
4. Congocidin BGC – Could the authors comment more exactly on Congocidine BGC transcription levels and boundary location? Is the boundary located within an operon? Is it separating different operons? If the latter, are these operons showing some kind of differential expression?
5. Stambomycin BGC – if transcription is responsible for boundary formation alone why does a boundary form at a lowly expressed BGC?
6. It may be useful to include a differential interaction analyses for the Hi-C maps obtained under different conditions.
7. BGCs can be found in both pro- and eukaryotes. It would be helpful to discuss the findings with respect to findings in eukaryotes. Similarly, a wider discussion of the observed correlations between chromosome architecture and transcriptional activity would be useful. E.g. Rowley et al. 2017 Mol Cell 67
8. L292-293. Comment not precise without corresponding metabolite data.
9. L313-314. Unclear how the authors back up this comment. This requires clarification. How do the authors define a transcriptional hub? Should this be similar to transcription factories? Furthermore, this concept is supported to only limited extend as (i) these boundaries also form in areas without high transcriptional activity and (ii) the boundaries are lost in the central area in exponential phase even though loss of transcriptional activity is very limited according to the presented data. Conformational data on any mutant with reduced transcriptional activity or additional experimental support via e.g. HiChIP or standard ChIP against a bacterial polymerase would significantly enhance these claims.

Ref: *Nature Communications* manuscript NCOMMS-21-01376-T

Manuscript: Dynamics of the compartmentalized *Streptomyces* chromosome during metabolic differentiation

Authors: Lioy *et al.*

Revision date: May 31st, 2021

Corresponding author: Stéphanie Bury-Moné

POINT-BY-POINT REPLY TO REVIEWER N° 1

Reviewer #1 stated:

In this study from the Bury-Moné Lab, Lioy et al investigated *Streptomyces* genome organization during metabolic differentiation. They analyzed the genome sequences of 125 species of *Streptomyces* and found that the center of the chromosome are enriched with core genes and the terminus region are enriched with metabolic genes. They performed transcriptome analyses and found that the metabolic genes are repressed during exponential growth and expressed during differentiation. Finally, they performed genome-wide chromosome conformation capture (3C-seq) experiments to investigate chromosome organization during exponential growth and differentiation.

The mechanism of chromosome organization and remodeling during differentiation remains an important and fundamental problem. It was previously reported that *Streptomyces* genomes are compartmentalized into central regions and terminal regions (Choulet 2006), and genes in these regions are differentially expressed in various growth conditions (Karoonthaisiri 2005, Jeong, 2016). Using 3C-seq, the authors found that the chromosome forms CIDs bounded by highly transcribed genes during exponential growth and the CIDs boundaries disappeared during development (which was essentially prolonged growth/stationary phase). This correlation of gene expression and CID boundaries were previously described in multiple bacterial species (Le 2013; Le 2016; Marbouty 2015; Lioy 2018; Bohm 2020, and more). Therefore, this manuscript confirms and extends previous findings but do not provide many new insights.

Our response: We thank Reviewer 1 for her/his evaluation of our work. We agree that “this manuscript confirms and extends previous findings” as written by reviewer #1. It is true that genetic compartmentalization was previously reported for the *S. ambofaciens* chromosome¹, while differential expression of the central and terminal regions was studied in *S. coelicolor*^{2,3} and studies on chromosome structure were performed in other genera (*Escherichia coli*, *Caulobacter crescentus*, *Bacillus subtilis*, *Pseudomonas aeruginosa*, *Corynebacterium glutamicum*,...). The originality of our study is the integration of several genomic approaches with new 3C data within a single species. Moreover, these analyses disclosed for the first time that compartmentalization of the *Streptomyces* genome correlates with gene synteny (hence genetic stability), gene expression in exponential/stationary phases as well as chromosome restructuring. No such compartmentalization with such functional outcomes has been described for other bacterial genomes.

To our knowledge, no previous study has integrated these three levels of analysis in the same study, which in itself brings originality to our approach. **For the first time, we show a link between genome evolution (compartmentalization, persistence), gene expression and chromosome structuring.**

Furthermore, to our knowledge, no study highlighted the fact that **the structural rearrangement of terminal regions coincides with specialized metabolite production** (note the new experiments below to expand on this point in response to Referee 3).

We have also developed new mathematical tools for 3C-seq analysis (such as the dispersion index) which indicate for the first time that two types of structuration simultaneously exist in *Streptomyces* chromosome (in the same growth condition), a phenomenon directly linked to the genetic and transcriptional compartmentalization.

In fact, the definition of the limits of the *Streptomyces* linear genome in terms of arms and core region remains challenging, since gene synteny gradually disappears towards the terminal ends. Thus, there is a need to establish a standardized nomenclature to compare the compartmentalization of *Streptomyces* genomes. Importantly our study highlights for the first time the need to examine the location of extreme ribosomal RNA (rRNA) operons in the delimitation of the large terminal compartments. This opens the possibility of formulating a new hypothesis concerning the limits of the compartmentalized genome of *Streptomyces* and the importance of these rRNA operons from an evolutionary point of view (see complementary information in the answer to Reviewer 2, p.6).

And then the reviewer stated:

It is interesting that the strength and extense of inter-arm contact increased during development compared to exponential phase. If they focused on the mechanism and biological relevance of this the change, the authors could have increased the impact of the work. There are some very straight forward experiments to address these. For example, are ParB/parS Smc-ScpAB required for the inter-arm contacts? Do cells have a higher level of these proteins during development? Is the increased inter-arm contact a consequence of the reduced transcription at stationary phase, i.e. if transcription is inhibited during exponential phase, would inter-arm contact increase? Does inter-arm contact have an impact on development, i.e. does losing inter-arm contact (in ParB/Smc mutants) have a negative effect on the development? Without these answers, this manuscript does not add much to the canon.

Our response: Reviewer #1 suggests some experiments, for instance the characterization of the activity of Smc-ScpAB during metabolic differentiation. Although this is an important question, this is not the objective of our study. Indeed, our work was submitted back to back with the complementary work of Szafran *et al.*⁴, in which the activity of Smc-ScpAB was specifically addressed during sporogenic development, with each of our papers delving into a different aspect of *Streptomyces*' cell cycle. Moreover, the activity of Smc-ScpAB in chromosome conformation and its interplay with ParB/parS is extensively described in the actinobacteria *Corynebacterium glutamicum*⁵ and in other bacterial models (*Caulobacter crescentus*⁶, *Bacillus subtilis*^{7,8}, *Pseudomonas aeruginosa*⁹). Interestingly, we did not observe an increase in the transcription of *smc-scpAB* during metabolic differentiation (Table S5) and Szafran *et al.*⁴ did not observe an increase of the protein level during sporulation. Thus, chromosome alignment by Smc-ScpAB must be regulated by an unknown mechanism. We agree with the reviewer that it would be very interesting to address the regulation of Smc-ScpAB activity during metabolic differentiation. However, this would be more appropriate in a separate, future study since it goes far beyond the scope of this report (Please also see the Response to Reviewer 3 p. 16, **Fig.R8**).

The following specific points were raised:

Minor points:

- 1) "Remodeling" indicates an active mechanism. However, the change of chromosome organization observed here could be passive, i.e. a byproduct of changed transcription. Without identifying a factor that remodels the chromosome, the authors should tune this down.

Our response: We agree with Reviewer 1 on the fact that the word 'remodeling' may suggest active changes in chromatin composition that we did not (yet) explore. Accordingly, we have replaced

‘remodeling’ by ‘rearrangement’ in the revised version of the manuscript. This is the same word used in the companion paper by Szafran *et al.*⁴.

2) Changing from “open” to “closed” conformation is an overstatement. In exponential phase (Fig 3A), there is a faint off-diagonal, so the chromosome is not really “open”, but is weakly closed.

Our response: Driven by this interesting comment from Reviewer 1, we applied a method to our data based on the filtering of significant 3C contacts developed by Wang *et al.*^{10,11} to quantify the extent of the secondary diagonal in *B. subtilis*. As illustrated in the figure below (**Fig. R1**), during exponential phase, the frequency of contacts is not significant enough to define a genuine secondary diagonal. Nevertheless, the significant contacts highlight a domain encompassing the origin of replication (**Figure R1**). Altogether, this suggests that in exponential phase, the overall conformation of the chromosome is rather open with increased interactions between loci located around the origin. We propose to replace in the text “open” to “rather open” and to add these figures and analysis in the revised version of the manuscript (legend of the **new Fig.5** and Supplementary **Figure 5.F**).

Figure R1: Analysis of the primary and secondary diagonals in the 3C-contact maps

Primary binary contact maps were obtained after setting the threshold for significant interactions at 0.5, 0.75 and 1 times the standard deviation (defined as $\sigma = 1.4826 * \text{mad}$ (median absolute deviation) above the median to the original 3C-contact maps (Fig. 3 & 4) obtained in exponential (C1 and C6 conditions *i.e.* 24 h in MP5 and YEME10 medium) and stationary phase (C4 condition *i.e.* 48 h in MP5 medium). Contact frequencies above or below the threshold were assigned a value of 1 or 0, respectively, generating a primary binary contact map in which significant interactions between chromosomal loci are represented in yellow, whereas non-significant interactions are represented in blue. Abscissa and ordinate axes represent genomic coordinates. Thereafter connected binary maps were generated by connecting 10 elements considered as significant (smaller elements are discarded as non-significant background noise) using a diamond shape of 15 to fill out the empty points comprised by the connected elements. Based on a method described in Wang *et al.*^{10,11}.

3) Figure 4F is not supported by the data. In exponential phase, there is some inter-arm contact in the data that is missing in the model. After 48 h of growth, only the regions near the origins are weakly interacting and these interactions do not extend to the entire genome. The authors should make the inter-arm contact weaker and shorter, not the strong alignment of the entire arms.

Our response: We built our model based on the 3D representation of the contact map using ShRec3D¹² already used in other bacteria^{8,13} and is presented in the figure below (**Fig. R2**). The resulting model is a

3D representation of the 2D contact map and does not represent the exact nucleoid structure found in an individual bacterium, but a convenient visualization tool of contact frequencies over a population of cells. Importantly, ShRec3D does not reconstruct 3D structures based on a polymer model, and it does not highlight the information concerning boundaries or gene expression. For this reason, we preferred to draw a model rather than to present only the 3D structure generated with ShRec3D.

As noted by Reviewer 1, there are increased contacts around the origin in exponential phase (a large domain rather than a secondary diagonal, represented by with the ‘V’ shape of the origin region). In stationary phase, the presence of a secondary diagonal indicates the high probability of the two arms of the chromosome contacting each other (also see response to point #2 above). We agree that the extent of contacts in the secondary diagonal decreased toward the terminal ends. We propose to emphasize this point by clearly stating in the legend of **Fig.5**: “Note that the inter-arms contacts observed in stationary phase are less frequent toward the terminal ends (**Supplementary Fig.5.F**)”.

We fully agree with Reviewer 1 about the need to improve the representation of the chromosome, and to include all this information. We therefore drew a new figure; closer to the 3D structures generated by ShRec3D, and overlaying the information about domains and boundaries identified in the contact maps. The new models as well as the 3D representations are presented below (**Fig. R2 & R3**) and are added to the revised version of the paper, replacing the previous model that we proposed.

Figure R2: 3D-models of *Streptomyces ambofaciens* genome generated by ShRec3D software
(New **Supplementary Fig.6.D** of the revised version of the manuscript)

Figure R3: Schematic representation of genome dynamics during metabolic differentiation

(New Fig. 5 of the revised version of the manuscript)

POINT-BY-POINT REPLY TO REVIEWER N° 2

Reviewer #2 stated:

Streptomyces are prolific antibiotic producers that are of great importance for society. About two-thirds of all antibiotics are derived from these bacteria. Unlike most other bacteria, streptomycetes have linear chromosomes, and many of gene clusters required for synthesis of these important metabolites are located on the so-called terminal ends of the chromosome. Despite decades of work, very little is known about the organisation of the linear chromosomes in these bacteria during growth. In this manuscript, the authors perform a very thorough investigation in the dynamics and organisation of the chromosome during differentiation. They use 3C-seq to demonstrate that the chromosome ends are relatively silent during primary growth, but become active during secondary metabolism whereby the chromosome also adopts a different conformation. In the latter state, the chromosome ends are more frequently in proximity of one another than during primary growth. The authors also show that the central part of the chromosome is organised in regions that are transcriptionally active and which are defined by so-called boundaries (such boundaries are also evident in "older cultures" when more active transcription is detected here). The manuscript is well written and presented. The work also provides important new insights that are interesting for a broad readership. It forms a foundational story that can lead to many interesting follow-up stories.

Our comment: We thank Reviewer 2 for her/his positive evaluation of our work and very good summary of the main outcomes of our study.

The following specific points were raised:

I have some suggestions to improve the manuscript.

1. Introduction/Results: The authors have chosen to do their work in *Streptomyces ambofaciens*. Although well-studied, this is not one of the canonical models in the field. Although this is of course not a problem, it would be good to describe in the introduction/beginning results why this model was chosen and not for instance *S. venezuelae* or *S. coelicolor*, which are more commonly used in the field.

Our response: We worked on *S. ambofaciens* ATCC23877 since: i) this strain is of biological interest and industrially exploited for the production of the macrolide spiramycin which is used in human medicine as an antibacterial and anti-toxoplasmosis agent, ii) we have extensive expertise in the genetic engineering of this strain including the sequencing of the genome by our groups¹⁴⁻¹⁶, and iii) this strain is well-studied with regard to genome organization and plasticity^{1,17-19}. We agree with Reviewer 2 that additional information concerning this strain should be included in the introduction. Accordingly, we add the following information:

Line 73: "In this work, we explore the dynamics of *Streptomyces ambofaciens* ATCC 23877 linear chromosome during metabolic differentiation. This strain, well-studied for its genome organization and plasticity, is industrially exploited for the production of spiramycin".

2. The arms of both chromosomes are quite different in length. Can the authors perhaps further explain how the disappearance of synteny (as described in Fig. 1, lines 104-105) relates to the different arms?

Our response: Indeed, Reviewer2 is correct when she/he points out that the two chromosomal arms are quite different in length. Relevant to this, by expanding our analysis to a panel of 124 other *Streptomyces* genomes, we found that the size imbalance between the terminal compartments observed in *S. ambofaciens* ATCC23877 is conserved in many *Streptomyces* species (**Figure R4** below).

Figure R4: Imbalance between terminal compartments in a panel of 125 *Streptomyces* genomes

Legend: The terminal compartments were defined with respect to the positions of the first and last operons encoding ribosomal RNAs (rDNAs) in a panel of 125 *Streptomyces* genomes. The sizes of the smallest ('MIN') and largest ('MAX') terminal compartment within each genome are presented. The MAX/MIN ratio has a mean of 1.9, and a median of 1.7 (standard deviation = 2.2). In 70 genomes (over 125), the right hand compartment is larger of the two arms, as in *S. ambofaciens* ATCC23877. In the other genomes, the left compartment is the largest.

Beyond the most extreme rDNAs, there are a few core genes and then both synteny and gene persistence decrease sharply (Fig.1, Fig. R5 below). Indeed synteny seems to diminish more rapidly in the shortest terminal compartment (which contains fewer genes from the core genome).

[Redacted]

In the discussion of the revised manuscript, we propose that the rDNA operons “constitute an evolutionary barrier that limits the occurrence of single recombination events within the central region” (line 368). In other words, we believe that the probability of the occurrence of recombination events that would lead to the loss of an rDNA operon is limited because of the risk of loss of fitness. This hypothesis is reinforced by the observation that the percentage of the core genome present between the two most extreme rDNAs varies little (remaining around 88.6 %) compared to the relative size of the central compartment or even of the region encompassing the entire core genome (Figure R6, below).

Figure R6: Percentage of core genes within the central compartment in a panel of 125 *Streptomyces* genomes (New Supplementary Fig.8)

Legend: The percentage of core CDSs located in the central region was assessed in the panel of 125 genomes representative of *Streptomyces* genus diversity. The sizes of the central compartment (delimited by the first and last rDNA operons) and of the region encompassing 100 % of the core CDSs ('complete core region') were also determined and expressed as percentages relative to the size of the whole chromosome (plasmids were not considered).

We plan to examine this idea further. Nevertheless, we think it is interesting to add already in the revised version of the manuscript the information concerning the part of the core genome present in the central region in order to support the hypothesis. We therefore propose to add **Fig. R6** as supplementary data, and the following sentence in the discussion:

Line 363: "Interestingly, the percentage of the core genome present between the two most extreme rDNAs varies little (remaining close to 88.6 %) compared to the relative size of the central compartment or even of the region encompassing the entire core genome (**Supplementary Fig. 8**)."

3. The use of Extended Data Figures and Supplementary Figures is confusing. I expect that this will be dealt with in the final version?

Our response: We fully agree with the Reviewer 2's comment. So we have completely reformatted the article so that there are only main or supplementary figures, which will make it easier to read. We thank Reviewer 2 for this good suggestion. Please find below the correspondence between the tables and figures of the initial and revised versions of the manuscript:

Title	Initial (submitted) version	Revised version
Main figures and table		
Growth conditions used to performed -omics analyses	Table 1	Table 1
The genetic compartmentalization of Streptomyces ambofaciens linear chromosome	Figure 1	Figure 1
Transcriptome dynamics depending on genome features and metabolic differentiation	Figure 2	Figure 2
Boxplot presenting the antisense index over growth in MP5 medium depending on feature of interest	Extended Data Fig.2.F	Figure 2.D
Spatial organization of Streptomyces ambofaciens chromosome and transcriptome in absence of metabolic differentiation	Figure 3	Figure 3
Chromosome rearrangement during metabolic differentiation	Figure 4A-E	Figure 4
Schematic representation of genome dynamics during metabolic differentiation	Figure 4F	Figure 5
Supplementary figures		
Size distribution of Streptomyces ambofaciens ATCC 23877 GIs	Supplementary Figure 1	Supplementary Figure 1
The metabolic differentiation of Streptomyces ambofaciens analyzed at the transcriptional level	Extended Data Figure 1 Panel A & B	Supplementary Figure 2 Panel A & B
HPLC analyses of the supernatants of Streptomyces ambofaciens ATCC 23877 over growth in MP5 medium	-	NEW - Supplementary Figure 2 Panel C
The metabolic differentiation of Streptomyces ambofaciens analyzed at the transcriptional level	Extended Data Figure 1 Panel C & D	Supplementary Figure 3
Transcriptomes over growth of the four biosynthetic gene clusters encoding all known antibacterial activities of Streptomyces ambofaciens ATCC 23877	Supplementary Figure 2	Supplementary Figure 3
The dynamic of Streptomyces ambofaciens transcriptome	Extended Data Figure 2 (except panel F)	Supplementary Figure 4
Mapping of sense- and antisense- reads along the chromosome over growth in MP5 medium	-	NEW -Supplementary Figure 4 F
Data exploration linking the chromosome architecture, transcription and gene persistence	Extended Data Figure 3	Supplementary Figure 5
Analysis of the primary and secondary diagonals in the 3C-contact maps	-	NEW -Supplementary Figure 5F
3C-contact-maps for Streptomyces ambofaciens grown in the studied conditions	Supplementary Figure 3 A,B,C	Supplementary Figure 6 A,B,C
3D-models of Streptomyces ambofaciens genome generated by ShRec3D software	-	NEW -Supplementary Figure 6.D
3C-contact-maps for Streptomyces ambofaciens grown in the studied conditions	Supplementary Figure 3 D	Supplementary Figure 6 E
Local rearrangement of the congocidine biosynthetic gene cluster induced by exogenous congocidine	-	NEW -Supplementary Figure 7
Percentage of core genes within the central compartment in a panel of 125 Streptomyces genomes	-	NEW -Supplementary Figure 8
Supplementary tables		
Collection of 125 Streptomyces genomes used in this study	Supplementary table 1	Supplementary table 1
List of the GIs identified in Streptomyces ambofaciens ATCC 23877 genome	Supplementary table 2	Supplementary table 2
Gene distribution in the central region versus the extremities (defined by the first and last rDNA operons) of Streptomyces ambofaciens ATCC 23877 chromosome depending on features of interest	Supplementary table 3	Supplementary table 3
List of the SMBGCs identified in Streptomyces ambofaciens ATCC 23877 genome	Supplementary table 4	Supplementary table 4
Overall results from the comparative genomics, RNA-seq and 3C-seq analyses	Supplementary table 5	Supplementary table 5

4. The authors seem to have used a modified version of YEME10 as mentioned in lines 388-390 (please indicate this clearly, as normal YEME10 contains 34% sucrose). Also, this medium is difficult to consider as a really biological relevant medium. Specifically, the medium imposes dramatic osmotic stress on the cells. How does this influence the organisation of the chromosome? Perhaps the authors can comment a bit on this in the discussion?

Our response: Yes, we used a modified version of the classic YEME medium with less sucrose to decrease the osmotic stress applied on the cells. To make this information clearer, as suggested by Reviewer 2, we modified the name of the medium in the article (replacing “YEME” by “YEME10”) and emphasized this point in the Methods section as follows:

Line 451: “a modified version of the YEME medium that we term “YEME10”, containing only 10.3 % sucrose”

We agree with the Reviewer that this medium is not relevant from a biological point of view to our knowledge. We used it essentially as a negative control (compared to the MP5 medium), because bacteria do significantly produce antibiotics in this condition. Another advantage of this medium (shared with the MP5 medium and quite few other media in our experience of growing our strain) concerns the fact that in this medium, *Streptomyces ambofaciens* grows in a rather dispersed way, which allows 3C-seq experiments to be performed.

Interestingly, the transcriptome of bacteria in the exponential growth phase varies only modestly between MP5 and YEME10 conditions (compared to other conditions, Revised **Supplementary Fig.3**). Huge differences are only observed in stationary phase (Revised **Supplementary Fig.3**). Indeed, the 3C contact maps obtained in exponential phase in MP5 and YEME10 media are quite similar (**Fig.3** and **Fig.4A**). We agree that the implication of these results could be more explicit. We therefore add the following sentence when presenting these results:

Line 222: "This result indicates both that the conformation of the chromosome is rather conserved in exponential phase regardless of the media composition, equally the osmotic pressure present in the YEME10 medium has relatively no impact on boundary formation."

5. I am curious if the authors think whether the change in chromosome conformation also relates to the genomic instability that is often witnessed in *Streptomyces*. The reason I mention this is because the authors seem to suggest that the high level of transcription in the central part of the chromosome, flanked by the boundaries is perhaps less prone to recombination (lines 318-320). Does this also mean that the recombination events (and genomic instability in the chromosomal arms) occurs during early growth (because those arms are not active then)?

Our response: Reviewer 2 raises here an excellent question that is at the heart of our working hypotheses. The possible existence of a link between chromosome structure and the notorious instability of the *Streptomyces* genome is one of the questions that emerge from our study. It is particularly interesting to note that in the exponential phase, the terminal domains indeed show different structural properties compared to the central part. At this stage, it is still difficult to give a definitive answer to this question. This requires the development of specific tools to study the rate of recombination along the chromosome, as a function of chromosome conformation and growth phase. This is undoubtedly a work that will be interesting to carry out in the future.

Minor comments

Our response: We are grateful to Reviewer 2 for taking the time to referee the manuscript carefully. All minor comments below have been addressed in the revised manuscript.

1. Line 29. Change to "...chromosome structure of *Streptomyces ambofaciens* correlates with..."

Our response: Changed

2. Line 46. The authors use e.g. However, what other forms of differentiation are meant? I think the authors mean i.e.?

Our response: Changed

3. Lines 60-64. The authors could include some references to work from other labs that have demonstrated genomic rearrangements (although admittedly the cited lab is leading in this respect).

Our response: We added the following references: ^{20,21}

4. Line 83-84 Change to: ...the numerous available *Streptomyces*"

Our response: Changed

5. Line 85. Omit "new" (it may be new for *Streptomyces*, but it is commonly used in other systems)

Our response: Changed

6. Line 115. Change "shorten" to "referred to as"

Our response: Changed

7. Line 120. Change to "terminal" (not terminals)

Our response: Changed

8. Line 138. remove "by *S. ambofaciens*"

Our response: Changed

9. Line 395. What filters were used?

Our response: We added in the revised version of the manuscript the reference to the filters we used (Whatman PVDF Mini-UniPrep syringeless filters, 0.2 μm).

10. Line 428. "owing to its location within the TIRs". This is not clear. Please rephrase

Our response: We rephrased as following: "since this SMBGC is located within the TIRs" (line 503).

11. Line 434. Change to "Identification of NAPSFs"

Our response: Changed

12. Line 448. $1e^{-10}$ 31?

Our response: Actually, "31" is the reference number. We agree that this is confusing. To clarify this point, we add "as previously described" in between $1e^{-10}$ and ³⁰ (line 523).

13. Line 452-458. Split into 2-3 sentences for clarity.

Our response: Changed

14. Line 802. Same legend than Figure 3?

Our response: Changed

15. Line 829. Change "under" to "in"

Our response: Changed

16. Line 830. Omit "although"

Our response: Changed

17. Line 841. I guess ATB is antibiotic?

Our response: Yes, ATB means 'antibiotic'. We removed the acronym from the whole manuscript and replaced it by "antibiotic".

18. Fig. 1. "Diversity" is without capital

Our response: Changed

19. Ext. Data Fig. 1. Why did the authors not include a 72h YEME10 sample? It seems that the halo size is quite a bit bigger than at 48 h in YEME10.

Our response: We presented the results obtained at 72 h to show that the observed difference in antibiotic production between MP5 and YEME10 media is not just a delay, but a significant defect. Yes, the halo size is slightly larger than at 48 h in YEME10, but the difference with production in MP5 medium is even greater at this growth time.

POINT-BY-POINT REPLY TO REVIEWER N° 3

Reviewer #3 stated:

In the manuscript under review, Lioy *et al* report an investigation into chromosome structure in *Streptomyces*. The authors focus their investigations on metabolic differentiation and highlight biosynthetic gene clusters. The organisation of genes related to specialised metabolism into clusters, groups of neighbouring functionally related genes, is a widespread phenomenon throughout both pro- and eukaryotic genomes. How chromosome folding in the three-dimensional cellular spaces relates to this organisational form is a largely untapped area of science. As such, the work presented here, represents a major milestone in bacterial genomics and *Streptomyces* biology. The first results section describes sequence compartmentalisation of the *S. ambofaciens* genome. These are largely standard analyses and confirm the expected pattern of gene distribution in *Streptomyces* genomes. The transcriptional analyses are powerful and clear. They underscore previous findings in *Streptomyces* on the transcriptional compartmentalisation of the chromosome. The 3Cseq analysis enters new territory in *Streptomyces* genomics. The authors show data for three different conditions (2x bacterial growth phase and 1x stationary phase) and correlate their Hi-C maps with the transcriptional analyses. The presented data reveals a chromosome compartmentalisation largely reflective of the sequence and transcriptional organisation. The central chromosomal region (transcriptionally active) is separated from the two chromosomal ends (transcriptionally inactive) during the growth phase. In addition, distinct domains are formed throughout the central region. Boundaries to these domains are enriched in transcriptionally highly active operons and highly conserved genes. During stationary phase, a new pattern of chromosome structure can be observed. The central chromosome region is not occupying a distinct domain anymore and instead novel boundary regions can be observed at two biosynthetic gene clusters. Furthermore, increased contacts between the chromosomal ends are detectable according to the 3Cseq data presented. Overall, an array of very solid data is presented in this manuscript. Data on genomics, transcriptomics and 3D chromosome structure are well-aligned to each other and the authors have been able to assemble a coherent story.

Our comment: We thank Reviewer 3 for her/his positive evaluation of our work as well as the interesting questions raised below.

However, I am struggling to connect metabolic differentiation with the presented data. There is no metabolite data shown and strong activation of a single BGC appears somehow limited evidence for metabolic differentiation. The authors claim activation of 20 % of putative SMBGC genes during stationary phase but it remains unclear to which extent this relates to 'real' metabolic changes. To me, chromosomal re-arrangements are clearly correlated to growth phase but, with the current data provided, only to limited extent to metabolic output.

Our response: In the article, we presented antibacterial bioassays (Revised **Supplementary Fig.2.B**), the pattern of transcription of the four SMBGCs encoding known antibacterial activities of *S. ambofaciens* (Revised **Supplementary Fig.3.C**) as well as the RNA-seq of all known or predicted SMBGCs (Supplementary Table S5). However, Reviewer 3 is correct in pointing out that we did not include metabolite data *per se*. Driven by this important remark, we propose to add to the supplementary data HPLC analysis of the culture medium after 24 h and 48 h of bacterial growth (**Fig. R7** below), and the following sentences in the Result and Method sections.

Figure R7: HPLC analyses of the supernatants of *Streptomyces ambofaciens* ATCC 23877 during growth in MP5 medium

Legend: After 24 h and 48 h of growth, filtered culture supernatants were submitted to HPLC analysis (see Method section for details). Sterile MP5 medium was used as a negative control. The absorbance was monitored at 297 nm. After 48 h of growth, the presence in the culture supernatants of several peaks (highlighted by numbers) that are absent after 24 h of growth illustrates the process of metabolic differentiation of *S. ambofaciens*. The peak #4 around 15.6 min presents the same retention time and absorbance spectrum absorbance (in the islet) than an authentic congocidine standard and is absent in the culture supernatant of a strain with the congocidine cluster deleted ('ΔCGC', lab collection). This peak therefore corresponds to congocidine. The results are representative of at least five independent experiments.

This figure is called in the text at this level:

Line 142: “Moreover, for *S. ambofaciens*, MP5 medium was previously reported to be suitable for the production of the antibiotics spiramycin²² and congocidine²³ while there is only limited antibiotic production in YEME10 medium (**Supplementary Fig.2**)”.

Line 333: “Moreover, we show that changes in gene expression dynamics are correlated with changes in the metabolome during differentiation (**Supplementary Fig.2**).”

Information added in the Method section:

Line 467: “HPLC analyses

After 24h ad 48 h of growth, supernatants were filtered through Mini-Uniprep syringless filter devices (0.2 μm, Whatman) and analyzed on an Atlantis dc18 100A (5 μm 4.6 mm x 150 mm) column

(temperature 31°C) using a Dioner Ultimate 3000 HPLC instrument (Thermo Scientific). Samples were 5-fold diluted in a solution of 20 µM cyclo(Trp-Trp). This latter was used for sample normalization since its retention time of 24.8 min is outside the window at which the detectable metabolite peaks produced by *S. ambofaciens* emerge. Samples were separated using the following method: 0 to 7 min , isocratic 0.1 % HCOOH in water (solvent A)/ 0.1 % HCOOH in CH₃CN (solvent B) (95:5) at 1 ml.min⁻¹ , 7 to 30 min, 5 % to 60 % solvent B, 30 to 35 min, 60 % to 100 % solvent B, 35 to 45 min, 100 % solvent B and 50 to 60 min, 5% solvent B. Metabolite production was detected at 297 nm.”

And then Reviewer #3 stated:

Furthermore, I see limited support presented here for a better understanding of BGC regulation. The authors provide two examples for an overlap between BGC location and boundary formation. Given that there seemingly are around 20 BGCs in the investigated species, the finding of just two boundary overlaps appears limited. One of the two examples shows very high expression and in the other example rather low expression. As such, it remains unclear how BGCs influence domain formation. The role of boundary formation in cluster regulation remains unclear as well (see specific question below).

Our response: We agree with Reviewer 3 that there is not a strict correlation between SMBGC transcription and the formation of boundaries. Actually, that is also the case in exponential phase for other type of genes expressed at very high level. An emblematic example is the case of the tmRNA (transfer messenger RNA, SAM23877_RS36805, 400 bp) which is the most abundant RNA (excluding ribosomal RNAs) in all studied conditions (Table S5). We did not observe any boundary associated to this gene. Indeed, the level of transcription does not seem to be the only criterion that governs the formation of a boundary, as previously observed in other bacteria^{6,8,13,24,25}. The boundaries are generally associated to highly expressed but also long operons, encoding membrane proteins for some of them. The congocidine BGC (29 kb) fulfills all these criteria which probably explains the fact that it forms such sharp boundaries in stationary phase. This cluster is indeed the largest SMBGC expressed at very high level at 48 h. We propose to add this information in the revised version of the manuscript.

Line 271: “The congocidine BGC (29 kb) is indeed the largest SMBGC expressed at very high level in this condition.”.

We agree that the second boundary observed in stationary phase, which corresponds to the largest SMBGC harbored by the strain i.e. the stambomycin BGC (146 kb), was not induced in the growth conditions we used. This suggests “the existence of other mechanisms, yet to be discovered that impose chromosome constraints” (as mentioned in the discussion of the submitted article, **line 371**). This phenomenon is not unprecedented since boundaries that fail to correlate with high gene transcription have also been detected in *B. subtilis*⁸. These boundaries correlate with the binding of Rok xenogeneic silencer. At this level, our study opens up avenues to explore that we hope will lead to the discovery of new actors in the regulation of SMBGCs. To be less evasive, we propose to add a more precise hypothesis in the Discussion, as followed:

Line 372: “Silent boundaries have also been described in *B. subtilis*⁸. They correlate with Rok xenogeneic silencer binding-sites⁸. Indeed, all *Streptomyces* possess two paralogs of the xenogeneic silencer Lsr2 which belong to both the core genome and the actinobacterial signature. One of them strongly represses SMBGC expression in *S. venezuelae*²⁶. Since Lsr2 is a bridging protein that can mediate DNA condensation into highly compact DNA conformation²⁷, its possible role in the formation of silent boundaries on very large SMBGCs (e.g. 146 kb for the stambomycin BGC) remains to be explored.”

Finally, to observe sharp boundaries, the local folding of the chromosome at the highly transcribed locus must be a feature of most of the bacterial genomes since 3C-seq is a cell population based technique.

Thus, heterogeneity in gene expression within the bacterial population may also mask the formation of boundaries occurring in a limited number of cells.

Therefore, we fully agree with Reviewer 3 that the causative link between boundary formation and cluster regulation is not established. Actually, going beyond the correlation remains a challenge in the field of chromosome conformation.

[Redacted]

And then Reviewer #3 stated:

As a pioneering study for *Streptomyces* genetics, the manuscript would be significantly strengthened if the authors could provide any type of confirmatory experiments to their 3C-seq chromosome architecture data. DNA-FISH is the gold standard in the community and appears feasible to present evidence for the different chromosomal condensation states in the tested conditions. If FISH prove to difficult to be carried out, it maybe possible to present locus specific 3C data. This would appear particularly useful for the Congocidin BGC.

Our response: We agree with Reviewer 3 that information about the chromosome disposition/folding inside cells would provide very valuable information. DNA FISH is a gold standard in eukaryotes, FROS (fluorescence repressor operator system) microscopy being more widely used in prokaryotes²⁸. Nevertheless, FROS requires genetic tools that we do not have developed yet for our model organism. Moreover, *Streptomyces ambofaciens* is a slow growing bacteria (\approx 1 week to obtain an isolated colony) whose genetic modification, although mastered in our laboratory, remains rather labor-intensive (need to introduce DNA by conjugation, design of genes with high GC content, limited number of genetic tools available compared to *E. coli*, etc...). Therefore, we cannot carry out such experiments within a reasonable time frame.

Instead, in **FigR9 (New Supplementary Fig.7)** and **revised Fig.4.F**, we present the analysis of the 3C-data concerning the congocidine CGC in exponential and stationary phase, as well as in presence of exogenous congocidine (new data). The following information was added to the manuscript:

Line 286: “To gain further insights on the link between expression and local rearrangement induced during metabolic differentiation, we further focused on the congocidine BGC. Interestingly, congocidine (also named ‘netropsin’) is a pyrrolamide binding the minor groove of the DNA double-helix in a sequence-specific manner (interaction at four or more consecutive A-T pairs)⁴¹. We took advantage of the fact that *S. ambofaciens* does not produce congocidine in exponential phase (especially in YEME10 medium), but can resist to this antibiotic, thus minimizing the impact of this compound on chromosome conformation at the global level. The addition of exogenous congocidine (1 μ g/ml) has no effect on the number and location of boundaries (**Supplementary Fig. 7.A**). Indeed, in this condition, the transcriptome varies little with only 5 genes significantly induced, all belonging to the congocidine BGC (**Supplementary Fig. 7.B**). The absence of an associated boundary highlights that the whole congocidine BGC should be expressed at very high level (as after 48 h growth in MP5) to observe a sharp boundary (**Supplementary Fig. 7.C**).”

Figure R9: Impact of exogenous congoicidine on congoicidine biosynthetic gene cluster architecture and expression

- A. **Contact map after 24 h in YEME10 growth medium in presence of exogenous congoicidine.** 3C-seq was performed on *S. ambifaciens* grown in exponential phase in YEME10 medium supplemented with 1 $\mu\text{g/ml}$ congoicidine. The color scale reflect the frequency of the contacts between genome loci, from white (rare contacts) to dark purple (frequent contacts).
- B. **Heatmaps of the congoicidine BGC transcription in different growth conditions.** Genes which expression is statistically regulated (compared to the control condition i.e. 24 h in the same growth medium) are boxed.
- C. **Focus on the genomic region (50 kb) encompassing the congoicidine cluster contact maps (square) in YEME10 growth conditions in absence (top) or presence (bottom) of exogenous congoicidine.** The focus was performed on the 3C-contact maps presented in Fig.3.A (YEME10) and in the panel A (YEME10 + congoicidine). The detailed organization of the congoicidine CGC is presented.

Then, the following specific points were raised:

Specific comments.

1. Fig 2a is unclear. Does each line represent a gene?

Our response: Yes, each line represents a gene. We added this information in the legend.

2. L 147 – expressed and/or regulated – unclear.

Our response: We agree that the sentence is unclear. We rephrase it this way:

Line 152: “We observed that the *S. ambofaciens* genome exhibits a transcriptional landscape with approximately 90 % of the genes significantly expressed in at least one condition (i.e. categorized in ‘CAT_1’ or higher, in at least one condition – Supplementary Fig.4.A & B, Supplementary Table 5). Moreover, this transcriptome is rather dynamic with more than 93 % of the genes differentially expressed in at least one condition (i.e. adjusted p value < 0.05 in the DESeq2 differential analysis using C1 condition as reference, Supplementary Table 5).”.

3. L 163-176 – antisense transcription – could the authors please expand. It is unclear how to align the positive correlation of antisense to sense transcription with the highest antisense index at lowly expressed genes. The authors suggest a role a gene regulation based on the presented data but avoid to outline a brief mechanistic model.

Our response: We agree with Reviewer 3 that we should clarify the case of antisense-transcription which is one of the highlights of this study. If we considered the absolute number of antisense reads, these are more abundant in genes expressed at very high level (in sense orientation), as illustrated below by mapping the sense and antisense reads along the chromosome (**Fig. R10**, new **Supplementary Figure 4.F panel**). Perhaps a state of the chromatin that is actively transcribing can explain this transcriptional ‘leakage’.

This observation favors the concept of ‘transcriptional hubs’, which we define as functional chromosome region enriched in transcription machineries.

Figure R10: Mapping of sense- and antisense- reads along the chromosome over growth in MP5 medium

The normalized counts [transcripts per million, normalized considering the total (sense- plus antisense-) transcription] were mapped on *S. ambofaciens* chromosome. Please note that the transcription in antisense orientation is presented at two different scales.

However, if we now analyze the level of antisense transcription relatively to the total transcription [what we called the ‘antisense index’, defined as the level of antisense-transcription over the total (sense plus antisense) transcription], we observe that the antisense index is very low in gene expressed at very high level (Revised **Fig.2D**). This suggests that the probability to initiate transcription in the sense-orientation (rather than the anti-sense one) is higher for genes expressed at very high level. At the opposite, poorly conserved regions (genomic islands, SMBGCs, plasmids, etc) present a high level of antisense transcription relatively to their transcription in sense orientation, especially in exponential phase

(Revised **Fig.2D**). This antisense transcription may contribute to lowering the expression of these regions. The antisense indices of GIs, pSAM1, prophage and especially SMBGCs tend to decrease in stationary phase (Revised **Fig.2D**), whereas it increases in the central compartment (Revised **Fig.2D, Fig. R10**). This suggests that the antisense transcription may be regulated and/or part of a regulatory process, as previously proposed³⁰. Altogether, our results highlight that the frequency of the transcriptional machinery recruitment as well as the probability of this machinery to initiate transcription in the correct orientation are key elements in the regulation of gene expression in *S. ambifaciens*.

These results suggest that high-level and specific gene expression indeed emerges from both an increase in the frequency of recruitment of the transcription machinery, and its correct location at gene promoter in sense orientation.

Driven by this comment, we propose to add the following sentences in the results and discussion of the revised version of the manuscript:

Line 179: “Highly transcribed genes promote an environment enriched in transcriptional machineries, which may generate a transcriptional ‘leakage’ in the antisense-orientation.”

Line 184: “This [antisense] index is low for genes expressed at very high level, suggesting that they have developed regulatory mechanisms that increase the probability of localizing properly the RNA polymerase at the promoter in the sense orientation.”

Line 193: “In this context, SMBGC switch-on may rely on mechanisms that limit antisense-transcription by increasing the probability of localizing properly the transcription machinery at promoters in the sense orientation³⁰.”

Line 338: “Pervasive transcription has been proposed to contribute to gene regulation and genome evolution³⁰. In this context, spurious transcription along the terminal arms may contribute to control gene expression over growth.”

4. Congocidin BGC – Could the authors comment more exactly on Congocidine BGC transcription levels and boundary location? Is the boundary located within an operon? Is it separating different operons? If the latter, are these operons showing some kind of differential expression?

Our response: The congocidine BGC is organized in eight transcriptional units (manuscript in preparation). In stationary phase, all genes are expressed at very high level (6.10^3 to $3.7.10^4$ normalized reads per kb after 48 h of growth, considering that the median number of reads per gene is 2.10^2 in the whole genome – Data available in **Supplementary Table S5**, Heatmaps added in **Fig. R9.B**). We observed a single boundary that covers the whole region (3 bins of 10 kb). This boundary may result from topological constraints imposed in this region by the intense transcription machinery activity that reduces interaction frequencies between the congocidine BGC and neighboring regions.

5. Stambomycin BGC – if transcription is responsible for boundary formation alone why does a boundary form at a lowly expressed BGC?

Our response: We have answered above (p. 15) to this question in our response to Reviewer 3’s first comments. Briefly, the precise mechanism leading to the formation of silent boundaries yet remain to be discovered. Maybe that the binding of Lsr2 DNA-bridging proteins (and/or other NAPs) to such a big SMBGC (146 kb) may be involved in this process, as Rok in *B. subtilis*⁸. This will require further investigations.

6. It may be useful to include a differential interaction analyses for the Hi-C maps obtained under different conditions.

Our response: Driven by Reviewer 3's suggestion, we generated the matrix presenting the ratio of 3C normalized maps obtained in MP5 medium (24 h versus 48 h, **Fig. R11**). As it can be observed, the heterogeneous nature of the multicellular population of *Streptomyces* in stationary phase makes difficult the comparison using this approach. Because of this, we decided to perform other types of analyses [frontier Index, variability index and statistical analyses of the matrices (Revised **Supplementary Figure 5**)], that allows a more reliable comparison of the different conditions studied.

Fig. R11: Ratio of normalized contact maps generated after 24 h and 48 h in MP5 growth medium.

For a given position in the map, a decrease or increase in contacts in stationary phase (compared to the exponential phase) is represented with a blue or a red signal, respectively.

7. BGCs can be found in both pro- and eukaryotes. It would be helpful to discuss the findings with respect to findings in eukaryotes. Similarly, a wider discussion of the observed correlations between chromosome architecture and transcriptional activity would be useful. *E.g.* Rowley *et al.* 2017 *Mol Cell* 67

Our response: Reviewer 3 is right in pointing out the value of including comparisons with processes occurring in eukaryotes. In fact, we began by mentioning the parallel between the genome compaction we observed and the transition from G1 to M during cell cycle (**line 415**). This parallel could be extended to the link between expression and dynamics of chromosome architecture. Accordingly, we add the following sentences and the proposed reference to the Discussion section.

Line 355: “Interestingly, transcription is also a major predictor of chromosome organization into small domains throughout Eukarya³¹”.

Line 399: “Interestingly, metabolic gene clusters have been reported to reside in dynamic chromosome 3D-domains sharing common transcriptional and epigenetic states in plants³² and probably in fungi³³. For instance, *Arabidopsis thaliana* BGC change their local folding upon expression, and are associated with silent chromatin when not expressed³². The relocalization of SMBGC away of a silent environment upon induction remains to be explored in bacteria.”

8. L292-293. Comment not precise without corresponding metabolite data.

Our response: We have now included HPLC analyses that illustrate the production of metabolites in stationary phase (please see **Fig. R7** and associated comments above). We added a call to this figure at the end of the sentence mentioned by Reviewer 3.

9. L313-314. Unclear how the authors back up this comment. This requires clarification. How do the authors define a transcriptional hub? Should this be similar to transcription factories? Furthermore, this concept is supported to only limited extent as (i) these boundaries also form in areas without high transcriptional activity and (ii) the boundaries are lost in the central area in exponential phase even though loss of transcriptional activity is very limited according to the presented data. Conformational data on any mutant with reduced transcriptional activity or additional experimental support via e.g. HiChIP or standard ChIP against a bacterial polymerase would significantly enhance these claims.

Our response: We thank Reviewer 3 for raising this interesting and challenging question. In fact, yes, we assume that transcriptional hubs may be similar to transcription factories i.e. microenvironment enriched in transcription machineries. In our study, almost all boundaries observed in exponential phase localize to persistent genes formed by long and highly expressed operons, with the notable exception of the boundary located at the origin of replication probably formed by the binding of ParB to *parS* sites as in *S. venezuelae* (cf. Szafran *et al.*, companion paper). These results indicate that transcription is closely linked to boundary formation in our model organism, as previously shown for other bacteria^{6,8,13,24,25}. We agree that the loss of boundaries in the central region, in stationary phase, is associated with only a limited decrease in the abundance of transcripts encoded by this region. The loss of boundaries may be either due to the loss of large operons expressed at extremely high level, and/or to the complete decrease in the transcription of genes from the central region (**lines 406 to 413**). Indeed, our RNA-seq analysis cannot distinguish between stable and nascent transcripts. Therefore, the second hypothesis cannot be excluded. Our preliminary results tend to confirm that relocating an SMBGC closed to the origin perturbs the chromosomal organization near *parS* sites (**Fig. R8**). ChIP against the RNA polymerase (and/or SMC complexes), as described in³¹, will definitely be required to investigate further this point.

REFERENCES

1. Choulet, F. *et al.* Evolution of the terminal regions of the *Streptomyces* linear chromosome. *Mol Biol Evol* **23**, 2361–9 (2006).
2. Karoonuthaisiri, N., Weaver, D., Huang, J., Cohen, S. N. & Kao, C. M. Regional organization of gene expression in *Streptomyces coelicolor*. *Gene* **353**, 53–66 (2005).
3. Jeong, Y. *et al.* The dynamic transcriptional and translational landscape of the model antibiotic producer *Streptomyces coelicolor* A3(2). *Nat Commun* **7**, 11605 (2016).
4. Szafran, Mj. *et al.* *Spatial rearrangement of the Streptomyces venezuelae linear chromosome during sporogenic development.* <http://biorxiv.org/lookup/doi/10.1101/2020.12.09.403915> (2020) doi:10.1101/2020.12.09.403915.
5. Böhm, K. *et al.* Chromosome organization by a conserved condensin-ParB system in the actinobacterium *Corynebacterium glutamicum*. *Nat Commun* **11**, 1485 (2020).
6. Le, T. B., Imakaev, M. V., Mirny, L. A. & Laub, M. T. High-resolution mapping of the spatial organization of a bacterial chromosome. *Science* **342**, 731–4 (2013).
7. Wang, X. *et al.* Condensin promotes the juxtaposition of DNA flanking its loading site in *Bacillus subtilis*. *Genes Dev* **29**, 1661–1675 (2015).
8. Marbouty, M. *et al.* Condensin- and Replication-Mediated Bacterial Chromosome Folding and Origin Condensation Revealed by Hi-C and Super-resolution Imaging. *Mol Cell* **59**, 588–602 (2015).
9. Lioy, V. S., Junier, I., Lagage, V., Vallet, I. & Boccard, F. Distinct Activities of Bacterial Condensins for Chromosome Management in *Pseudomonas aeruginosa*. *Cell Reports* **33**, 108344 (2020).
10. Wang, X. *et al.* In Vivo Evidence for ATPase-Dependent DNA Translocation by the *Bacillus subtilis* SMC Condensin Complex. *Mol Cell* **71**, 841-847.e5 (2018).
11. Wang, X., Brandão, H. B., Le, T. B. K., Laub, M. T. & Rudner, D. Z. *Bacillus subtilis* SMC complexes juxtapose chromosome arms as they travel from origin to terminus. *Science* **355**, 524–527 (2017).

12. Lesne, A., Riposo, J., Roger, P., Cournac, A. & Mozziconacci, J. 3D genome reconstruction from chromosomal contacts. *Nat Methods* **11**, 1141–1143 (2014).
13. Liou, V. S. *et al.* Multiscale Structuring of the E. coli Chromosome by Nucleoid-Associated and Condensin Proteins. *Cell* **172**, 771-783 e18 (2018).
14. Aigle, B. *et al.* Genome mining of *Streptomyces ambofaciens*. *J Ind Microbiol Biotechnol* **41**, 251–63 (2014).
15. Najah, S., Saulnier, C., Pernodet, J. L. & Bury-Mone, S. Design of a generic CRISPR-Cas9 approach using the same sgRNA to perform gene editing at distinct loci. *BMC Biotechnol* **19**, 18 (2019).
16. Thibessard, A. *et al.* Complete genome sequence of *Streptomyces ambofaciens* ATCC 23877, the spiramycin producer. *J Biotechnol* **214**, 117–8 (2015).
17. Hoff, G., Bertrand, C., Piotrowski, E., Thibessard, A. & Leblond, P. Genome plasticity is governed by double strand break DNA repair in *Streptomyces*. *Sci Rep* **8**, 5272 (2018).
18. Tidjani, A. R. *et al.* Massive Gene Flux Drives Genome Diversity between Sympatric *Streptomyces* Conspecifics. *MBio* **10**, (2019).
19. Lorenzi, J.-N., Lespinet, O., Leblond, P. & Thibessard, A. Subtelomeres are fast-evolving regions of the *Streptomyces* linear chromosome. *Microbial Genomics* (2021) doi:10.1099/mgen.0.000525.
20. Hopwood, D. A. Soil to genomics: the *Streptomyces* chromosome. *Annu Rev Genet* **40**, 1–23 (2006).
21. Zhang, Z. *et al.* Antibiotic production in *Streptomyces* is organized by a division of labor through terminal genomic differentiation. *Sci Adv* **6**, eaay5781 (2020).
22. Pernodet, J. L., Alegre, M. T., Blondelet-Rouault, M. H. & Guerineau, M. Resistance to spiramycin in *Streptomyces ambofaciens*, the producer organism, involves at least two different mechanisms. *J Gen Microbiol* **139**, 1003–11 (1993).
23. Juguët, M. *et al.* An iterative nonribosomal peptide synthetase assembles the pyrrole-amide antibiotic congocidine in *Streptomyces ambofaciens*. *Chem Biol* **16**, 421–31 (2009).

24. Le, T. B. & Laub, M. T. Transcription rate and transcript length drive formation of chromosomal interaction domain boundaries. *EMBO J* **35**, 1582–95 (2016).
25. Val, M. E. *et al.* A checkpoint control orchestrates the replication of the two chromosomes of *Vibrio cholerae*. *Sci Adv* **2**, e1501914 (2016).
26. Gehrke, E. J. *et al.* Silencing cryptic specialized metabolism in *Streptomyces* by the nucleoid-associated protein Lsr2. *Elife* **8**, (2019).
27. Qu, Y., Lim, C. J., Whang, Y. R., Liu, J. & Yan, J. Mechanism of DNA organization by *Mycobacterium tuberculosis* protein Lsr2. *Nucleic Acids Res* **41**, 5263–5272 (2013).
28. Le, T. B. & Laub, M. T. New approaches to understanding the spatial organization of bacterial genomes. *Curr Opin Microbiol* **22**, 15–21 (2014).
29. Neidle, S. DNA minor-groove recognition by small molecules (up to 2000). *Nat. Prod. Rep.* **18**, 291–309 (2001).
30. Wade, J. T. & Grainger, D. C. Pervasive transcription: illuminating the dark matter of bacterial transcriptomes. *Nat Rev Microbiol* **12**, 647–653 (2014).
31. Rowley, M. J. *et al.* Evolutionarily Conserved Principles Predict 3D Chromatin Organization. *Mol Cell* **67**, 837-852.e7 (2017).
32. Nützmänn, H.-W. *et al.* Active and repressed biosynthetic gene clusters have spatially distinct chromosome states. *Proc Natl Acad Sci U S A* **117**, 13800–13809 (2020).
33. Winter, D. J. *et al.* Repeat elements organise 3D genome structure and mediate transcription in the filamentous fungus *Epichloë festucae*. *PLoS Genet* **14**, e1007467 (2018).

REVIEWERS' COMMENTS

Reviewer #1 (Remarks to the Author):

The authors have done a great job addressing my concerns with new figures. The manuscript is improved by the revision.

Reviewer #2 (Remarks to the Author):

The authors have done a good job at addressing the concerns raised by me, but also the other referees. I want to congratulate the authors with this great story.

Reviewer #3 (Remarks to the Author):

Overall, the revised manuscript addresses all major points adequately. The new data presented in R8 looks particularly exciting and I am looking forward to reading about this in a future manuscript.